# The release of toxic oligomers from α-synuclein fibrils induces dysfunction in neuronal cells

Roberta Cascella [1], Serene W. Chen [2,3], Alessandra Bigi [1], José D. Camino [4], Catherine K. Xu [3], Christopher M. Dobson [3], Fabrizio Chiti [1], Nunilo Cremades [4✉] & Cristina Cecchi [1✉]

The self-assembly of α-synuclein (αS) into intraneuronal inclusion bodies is a key characteristic of Parkinson's disease. To define the nature of the species giving rise to neuronal damage, we have investigated the mechanism of action of the main αS populations that have been observed to form progressively during fibril growth. The αS fibrils release soluble prefibrillar oligomeric species with cross-β structure and solvent-exposed hydrophobic clusters. αS prefibrillar oligomers are efficient in crossing and permeabilize neuronal membranes, causing cellular insults. Short fibrils are more neurotoxic than long fibrils due to the higher proportion of fibrillar ends, resulting in a rapid release of oligomers. The kinetics of released αS oligomers match the observed kinetics of toxicity in cellular systems. In addition to previous evidence that αS fibrils can spread in different brain areas, our in vitro results reveal that αS fibrils can also release oligomeric species responsible for an immediate dysfunction of the neurons in the vicinity of these species.

[1] Department of Experimental and Clinical Biomedical Sciences, Section of Biochemistry, University of Florence, Florence, Italy. [2] Department of Life Science, Imperial College London, London, UK. [3] Centre for Misfolding Diseases, Department of Chemistry, University of Cambridge, Cambridge, UK. [4] Institute for Biocomputation and Physics of Complex Systems (BIFI), Joint Unit BIFI-Institute of Physical Chemistry "Rocasolano" (CSIC), University of Zaragoza, Zaragoza, Spain. ✉email: ncc@unizar.es; cristina.cecchi@unifi.it

Intracellular fibrillar inclusions of the protein α-synuclein (αS), called Lewy bodies, are characteristic of a range of neurodegenerative disorders, collectively referred to as synucleinopathies[1]. Most of the experimental evidence indicates that protein aggregation is a gain-of-toxic-function process that plays a central role in the pathogenesis[2,3]. A variety of in vitro studies is indicative of a nucleated conformational conversion model of αS, in which the intermediate oligomeric species are inherently transient, heterogeneous, and only populated at low levels during aggregation[4–7].

Growing experimental evidence suggests that specific oligomeric species are the most cytotoxic forms of αS and play a key role in disease[1,4,6,8–11]. A detailed characterization of two structural types of stable αS oligomers that were trapped using different strategies was recently reported[4,8]. One type is largely disordered in terms of secondary structure and the other type contains a β-sheet rich core; we refer to these species as type-A* oligomers (OA*) and type-B* prefibrillar oligomers (OB*), respectively[8]. Interestingly, we found that while OA* are able to interact with the membrane surface, although in an unspecific manner and with no apparent membrane insertion and perturbation, prefibrillar OB* were found not only to establish strong interactions with the bilayer through its unstructured N-terminal segment (residues 1–26), but also to insert its β-sheet core in the interior of the bilayer, causing major membrane disruption[4,8,12,13]. This causes only OB* to be toxic, whereas OA* are biologically inert[8,12].

On the other hand, αS fibrils have also been reported to be toxic and their toxicity has been associated with membrane perturbation[14–16]. Evidence is also accumulating that certain αS species transmit and induce seeding in neighboring cells in a spreading process in which different areas of the brain are slowly and inexorably affected[17–21]. Uptake of small αS fibrillar species by neuronal cells has indeed been observed[22,23] and particularly small αS fibrils are indeed used to induce Parkinson's disease (PD)-like phenotypes when injected in mice brains[18,23,24]. In order to have a thorough and exhaustive understanding of pathogenesis in PD and other synucleinopathies, the effect of fibril spreading, fibril toxicity, and oligomer toxicity have to be considered.

Here we have established the mechanisms of membrane perturbation and dysfunction to neuronal cells of highly stable and well-defined αS aggregates, such as type-B* prefibrillar oligomers (OB*), short fibrils (SF), and long fibrils (LF) under physiological conditions and endogenous levels of αS expression. In addition to the well-established contribution of αS fibrils to the diffusion of the pathology by a spreading mechanism, our results show that the fibrillar species can have an immediate toxic effect due to the release of toxic oligomeric species in human iPSC-derived dopaminergic neurons, rat primary cortical neurons, and human SH-SY5Y neuroblastoma cells. These findings reveal that the αS species detected in neurons and SH-SY5Y cells by confocal and stimulated emission depletion (STED) microscopy using different probes arise from exogenous human αS derived from SF/LF, rather than endogenous rat αS converted in oligomeric form by a seeding process.

## Results

### αS fibrils interact weakly with the surface of the lipid bilayer of synthetic membranes.
All species studied in this work (OB*, SF, LF) were first characterized for morphological, structural, and tinctorial properties with atomic force microscopy (AFM), far-UV circular dichroism (CD), and Fourier-transform infrared (FT-IR) spectroscopy, X-ray diffraction, and thioflavin T (ThT) and 8-anilinonaphthalene-1-sulfonic acid (ANS) fluorescence (Table S1, Figures S1, and S2a-e). We also characterized the monomer (M) and OA* as negative controls. This allowed their

identity to be confirmed, relative to previous works, and their properties to be studied. In brief, OA*/OB* are globular-like aggregates with height of 4–5 nm, whereas SF and LF have elongated morphologies with similar height but lengths differing by one and two orders of magnitude with respect to height, respectively. OA*/M show disordered secondary structure. SF/LF possess similarly high degrees of β-sheet conformation, whereas OB* species have an intermediate content. Unlike OA*/M, OB* have a significant but weak ThT binding, whereas SF/LF have a large ThT binding. Unlike M/OA*, OB*/SF/LF also have a strong ANS binding, indicating a high degree of solvent-exposed hydrophobicity.

Using paramagnetic relaxation enhancement (PRE) with magic angle spinning (MAS) solid-state nuclear magnetic resonance (ssNMR) spectroscopy, it was recently shown that the prefibrillar OB* species are able to anchor to synthetic synaptic-like small unilamellar vesicles (SUVs), through binding of the exposed N-terminal region of αS, and then to penetrate into the bilayer with its structured β-sheet core, causing substantial perturbations to its structure[8]. In the same study it was shown that M/OA* interact with the membrane surface, but without insertion into the lipid bilayer[8,25]. In the present work, we extended such studies to the SF and LF fibrillar species. We probed, in particular, the interaction of the rigid cores of the SF and LF (140 μM monomer equivalents) with SUVs with DOPE:DOPS:DOPC lipid composition (molar ratio of 5:3:2). In these experiments, small quantities of lipid molecules labeled with a paramagnetic center (PC), that induces relaxation in the nucleus of nearby atoms, were incorporated into the bilayers[8,26]. Unlike prefibrillar OB*, we did not observe any selective quenching of resonances in the $^{13}C$–$^{13}C$ dipolar-assisted rotational resonance (DARR) correlation spectra of either SF or LF, when the PC was located either in the external hydrophilic head groups (Fig. 1a, left panels) or the internal hydrophobic tails of the lipid molecules (Fig. 1a, right panels). This analysis indicates that, in contrast to OB*, the rigid β-sheet cores of neither SF or LF are able to interact significantly with the lipid bilayer. Nor do they appear to interact with the surface of the bilayer, as the core of the OA* species does.

To further study the interaction of the OB*/SF/LF species with the lipid bilayer with a different technique, we then analyzed the abilities of these species (10 μM monomer equivalents), to interact and bind the same SUVs, using far-UV CD as a spectroscopic probe. In each case, the spectrum of the aggregates without SUVs was subtracted from that acquired in their presence, so that the spectrum of the region of the protein interacting with SUVs could be obtained (Fig. 1b). While the acquisition of α-helical structure by M results in large changes in its CD spectrum upon interaction with lipids[25], the observed spectral changes were progressively smaller for OB*, SF, and LF (Figure S2f). The changes in the CD spectra of the fibrils cannot be attributed only to the acquisition of helical structure (as in OB*), but also to the appearance of random coil structure (Fig. 1b). Overall, the far-UV CD data indicate that the OB*, SF, and LF are able to interact with SUVs but with a progressively decreasing affinity, and that the interaction in each case results in the acquisition of some helical structure, likely to be in the unstructured and MAS ssNMR-invisible N-terminal region of the protein[25], which is readily accessible in OB* but significantly less accessible in the fibrils.

To analyze whether or not the differences in the interaction of the prefibrillar OB* and the fibrils with lipid vesicles are reflected in differences in their ability to disrupt membrane integrity, we compared their rates and extent of induced calcein release using SUVs composed of POPS lipids containing calcein molecules trapped in their interior. We found that OB* induced a rapid and substantial calcein release, with ca. 30%

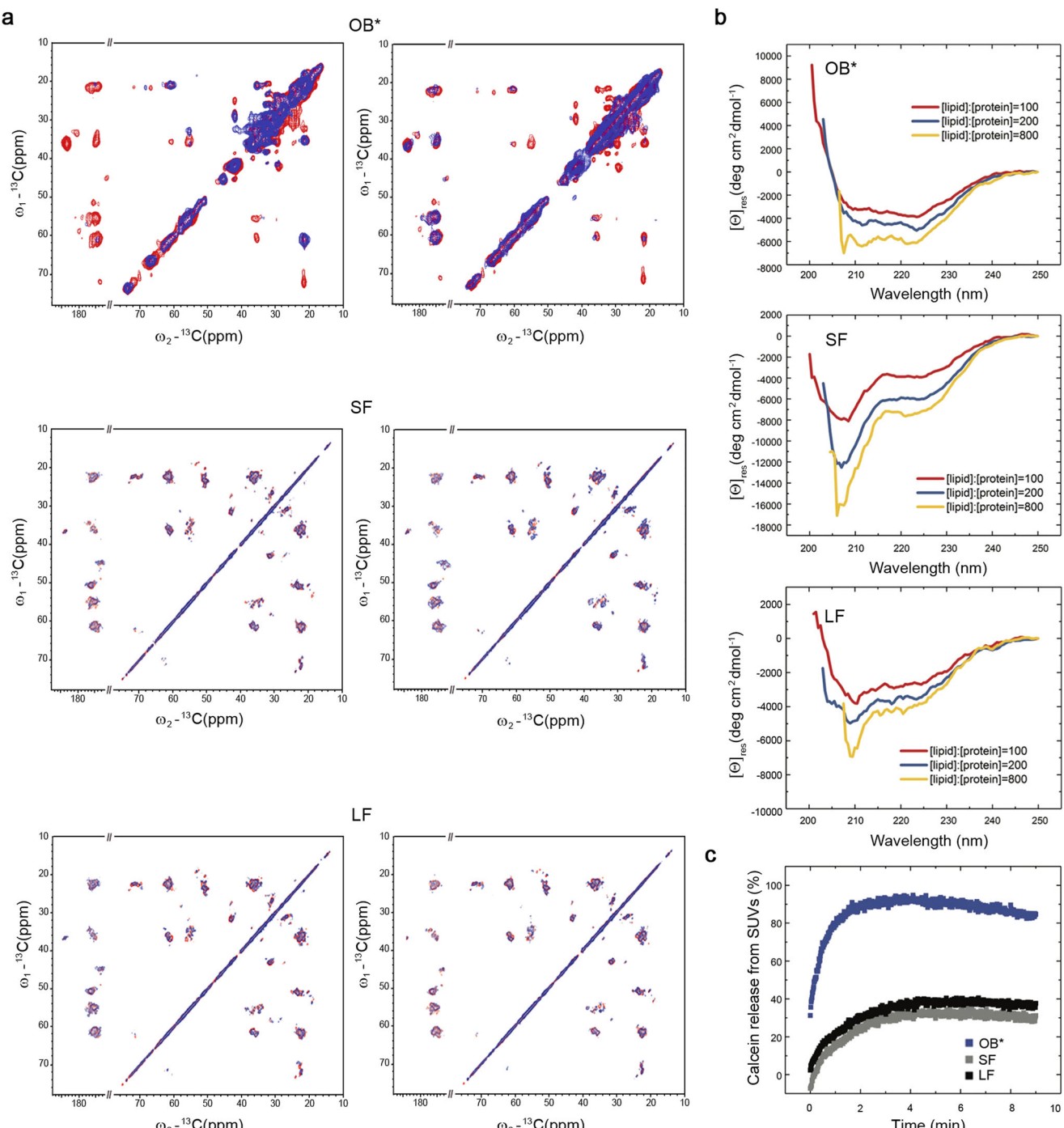

**Fig. 1 αS fibrils interact weakly with the surface of the lipid bilayer of synthetic membranes. a** PRE effects measured using MAS ssNMR for OB*/SF/LF using SUVs with DOPE:DOPS:DOPC lipid composition (molar ratio of 5:3:2) with a paramagnetic center (PC) on the bilayer surface on the hydrophilic head group (left) or in the membrane interior at carbon 16 of the lipid chain (right). $^{13}$C–$^{13}$C DARR spectra measured in the presence and absence of the PC-labeled lipids are shown in blue and red, respectively. For comparison, the plots of OB* are drawn using OB* data from our previous investigation[8]. **b** Changes in the far-UV CD spectrum of 10 μM OB*/SF/LF and increasing concentrations of SUVs: 1 mM (red), 2 mM (blue), and 8 mM (yellow). The spectra in the absence of SUVs was subtracted in each case from that acquired in their presence. **c** Calcein release from SUVs (% of total intravescicular calcein—signal normalized with respect to the treatment with 1% v/v Triton X-100, see Supplementary Information for more details) upon incubation of the vesicles with the indicated αS species.

of maximum release at the first time point of measurement, and reaching the maximum level (ca. 90% of maximum release) after 2 min of incubation at a protein:lipid ratio of 1:100. SF/LF caused, however, a much slower and very significantly reduced calcein release (ca. 30% release after 5 min) under identical conditions (Fig. 1c).

Taken together, these findings indicate that fibrils can interact with the membrane surface likely through the N-terminal region of the protein, but do not cause significant disruption of the lipid bilayers as they cannot insert their β-sheet core, whereas prefibrillar OB* are particularly effective in disrupting membrane integrity by inserting their partially-formed β-sheet core.

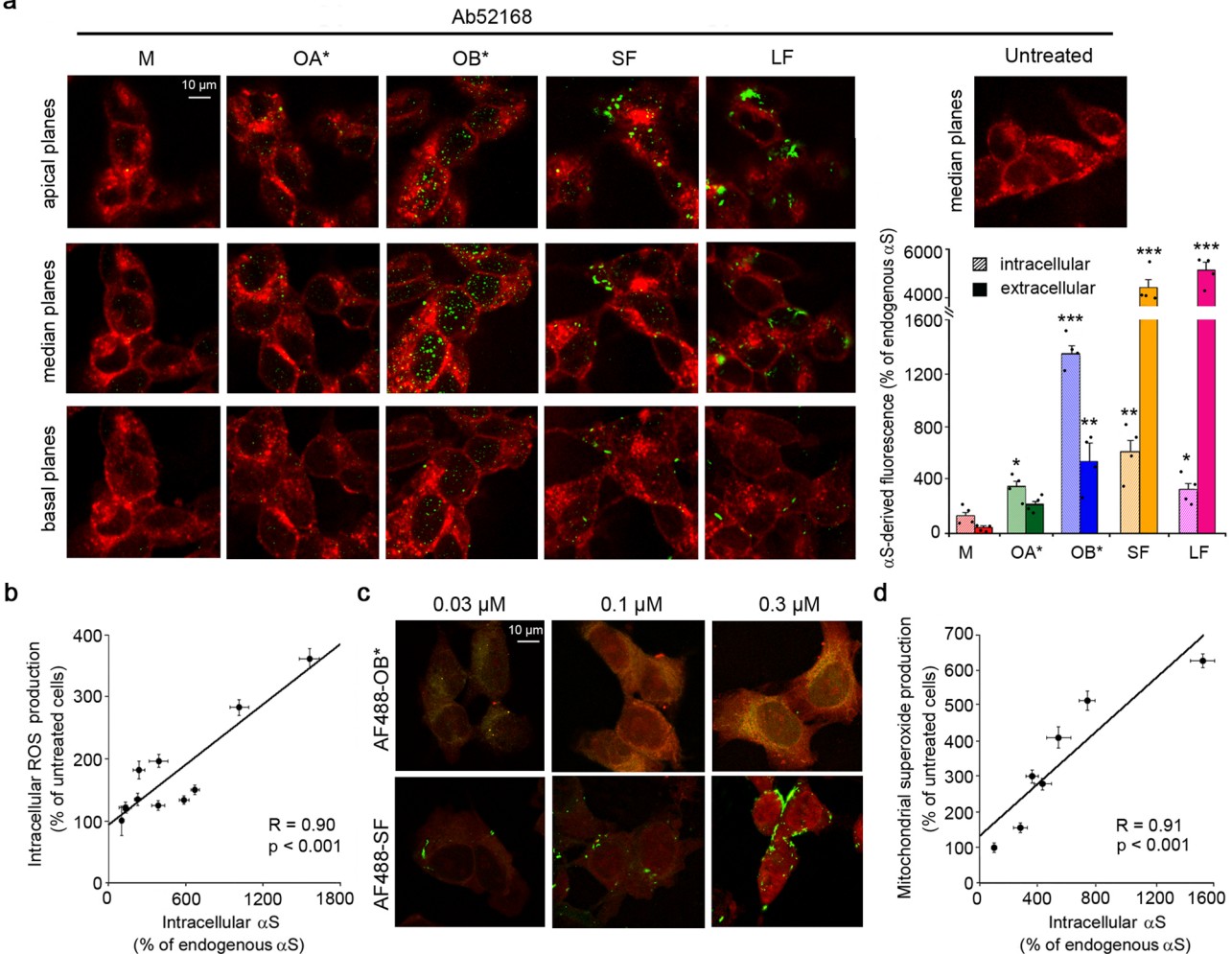

**Fig. 2 αS fibrils are largely localized at the surface of the cellular membrane but ROS generation correlates with the intracellular αS pool. a**
Representative confocal scanning microscope images showing the basal, median, and apical sections of SH-SY5Y cells treated for 1 h with the indicated αS species at 0.3 µM and the median sections of untreated cells. Red and green fluorescence indicates the cell membranes and the αS species revealed with wheat germ agglutinin (WGA) and polyclonal anti-αS antibodies (Ab52168, Abcam), respectively. The histogram on the right reports a semi-quantitative analysis of the intracellular and extracellular αS-derived fluorescence data expressed as the percentage of endogenous αS fluorescence. **b** Dependence of ROS production on the intracellular αS-derived fluorescences in SH-SY5Y cells treated with αS species. ROS values reported in Fig. S5 were plotted against the αS-derived fluorescence values reported in Fig. S4d of cells treated with OB* and SF at the corresponding concentrations. **c** Representative confocal scanning microscope images showing mitochondrial superoxide production detected with MitoSOX probe in living SH-SY5Y cells. Red and green fluorescence indicates MitoSOX staining and αS labeled with AF488 dye, respectively (six independent experiments with one internal replicate).
**d** Dependence of mitochondrial superoxide production on the intracellular AF488-derived fluorescence signal in SH-SY5Y cells treated with αS species. Experimental errors are S.E.M. (n = 4 with three internal replicates in panels (**a**), (**b**); n = 6 in panel (**d**) with one internal replicate). Samples were analyzed by one-way ANOVA followed by Bonferroni's multiple comparison test relative to untreated cells (in panel **a**, *P < 0.05, **P < 0.01, ***P < 0.001; in panel **b**, P < 0.001; in panel **d**, P < 0.001). A total of 200–250 cells were analyzed per condition.

**αS fibrils are largely localized at the surface of the cellular membrane**. To improve our understanding of the molecular mechanisms underlying PD pathogenesis and progression, we have started our experiments on SH-SY5Y neuroblastoma cells because of their human origin and because these cells express tyrosine hydroxylase (TH), dopamine-beta-hydroxylase, and the dopamine transporter, thus recapitulating many characteristics of human dopaminergic neurons. We initially investigated the ability of OB*/SF/LF (0.3 µM monomer equivalents) to interact with the plasma membrane and penetrate into the cytosol of SH-SY5Y cells, when incubated in the cell medium (CM) for 1 h. The αS species (green channel) were counterstained and analyzed by confocal scanning microscopy at the cellular apical, median, and basal planes of the plasma membrane (red channel) parallel to the

coverslip (Fig. 2a), to make a clearer distinction between intra-cellular vs extracellular αS pools[8]. M and OA* interacted weakly with the cellular membranes and hardly entered into the cells, whereas OB* and, to a lesser extent SF, were effectively able to cross the cellular membrane (Fig. 2a). Higher intracellular protein levels (median planes) were measured after 1 h of treatment with OB* with respect to SF and LF (1377 ± 62, 624 ± 106% and 322 ± 54%, respectively) relative to the very low signal of endogenous αS in untreated cells, taken as 100% (Fig. 2a). Similar results were obtained from the semi-quantitative analysis of the intracellular number of αS puncta (Figure S3a). Notably, a mild protease treatment with 0.05% trypsin at 4 °C, that digested all the protein molecules exposed on the cell membrane, including the surface-bound OB*, drastically reduced the green-fluorescent signal at the

apical planes (Figure S3b), without any significant modification of intracellular αS aggregates at the median planes of SH-SY5Y cells (Figure S3b). These results indicate that the signal observed at median planes arose, indeed, from the internalized OB*. Moreover, inhibition of endocytosis, either by a pharmacological treatment with dynasore or by a temperature-mediated blockage, significantly decreased, but did not cancel, the amount of intracellular aggregates in treated cells (Figure S3b), indicating that the influx of OB* through the membrane bilayer is attributable to a mixture of passive diffusion and endocytosis.

The ratio between the protein signal at the median planes and the total protein signal from all planes revealed that ~71%, 13% and 6% of the OB*, SF, and LF were found to penetrate the cells after 1 h of incubation, respectively (Fig. 2a). Thus, most of the αS-derived signal was localized at the plasma membrane in cells exposed to SF/LF, respectively.

When SH-SY5Y cells were exposed to OB*, SF, and LF for longer time periods (3, 6, and 24 h) a progressive and significant increase in intracellular species at cellular median planes was observed, although in cells treated with SF/LF the intracellular pool remained smaller than the extracellular one, even after 24 h, while for OB* ca. 90% of the protein is intracellular at already 3 h of incubation (Figure S3c).

A dose-dependent analysis revealed that, for both OB* and SF, similar effects were observed at lower αS concentrations with a clear dose-dependence, although at higher αS concentrations, no significant increase in intracellular αS levels was detected (Figure S4a,b,d), likely as a result of the agglutination of aggregated αS species at high concentrations[27]. Moreover, OB* showed a similar bell-shaped protein concentration dependence for both the intracellular and the total αS-derived fluorescence with the amount of the oligomers inside the cells found to be directly proportional to the mass concentration of oligomers interacting with the plasma membrane (Figure S4d,e). By contrast, the total amount of SF bound to the membrane was much higher than the intracellular fraction at any αS concentration (Figure S4b,d,e). Furthermore, concentrations higher than 0.3 µM resulted in an increase in the number of fibrils bound to the cell membrane but not in the quantity of intracellular protein (Figure S4b,d,e). These results indicate that the extent to which OB*, though not SF, has been taken up by the cells correlates with the amount of αS interacting with the plasma membrane.

**ROS generation correlates with the intracellular αS pool**. The generation of intracellular reactive oxygen species (ROS) is one of the inflammatory sequelae associated with protein aggregate treatment[28] and one of the earliest biochemical changes experienced by cells exposed to pathogenic αS aggregates[6]. We have also previously found that OB*, and to a lesser extent SF, induce a significant increase in ROS levels in SH-SY5Y cells, whereas OA* and M were found to cause a non-significant effect[8]. Here we extended the measurements to various αS concentrations of OB*/SF/LF, with the aim to relate a readout of toxicity with that of αS penetration and cell surface binding. ROS production showed again a bell-shaped αS concentration dependence when the different αS species were added for 15 min to the CM of SH-SY5Y cells, with a maximum effect at 0.3 µM (Figure S5a,b). A plot of the degree of the ROS increase, measured at the different concentrations of OB*/SF, against the intracellular αS levels, using the fluorescent signals obtained for OB*/SF, revealed a very significant positive correlation (R = 0.90, $p < 0.001$) (Fig. 2b). By contrast, no significant correlation was found between ROS increase and total αS, i.e., sum of intracellular and extracellular pools (R = 0.046, $p > 0.05$; Figure S5c).

We also monitored concomitantly the generation of mitochondrial superoxide ions and intracellular αS levels in living SH-SY5Y cells exposed for 1 h to 0.03, 0.1, and 0.3 µM of OB*/SF labeled with AF488 dye or to 250 µM $H_2O_2$ as a positive control (Fig. 2c and S5d). The increase in intracellular αS levels (green channel) correlated positively with the mitochondrial superoxide production (red channel) in a dose-dependent relationship (R = 0.91, $p < 0.001$) (Figs. 2c, d and S5d). These results indicate that the generation of ROS and mitochondrial superoxide is related to the level of intracellular αS.

**αS fibrils gradually destabilize membrane integrity resulting in neuronal dysfunction**. In order to confirm the ability of the different αS species to destabilize the membrane permeability of cells, we assessed the release of the fluorescent probe calcein-AM, previously loaded into the cells[29]. Calcein acetoxymethyl (Calcein-AM) is a substrate that passively crosses the cell membrane and in the cytosol is hydrolyzed by the enzyme esterase to a polar green-fluorescent product (calcein) that is retained into cells with intact membrane. The addition to the CM for 1 h of 0.3 µM OB* and, to a lesser extent, SF, generated a significant leakage of calcein, such as $73 \pm 2\%$ and $41 \pm 3\%$ in rat primary cortical neurons and $64 \pm 4\%$ and $37 \pm 2\%$ in human SH-SY5Y cells, respectively (Fig. 3a). Calcein leakage suggests a permanent disruption of neuronal bilayers following the aggregates penetration through membranes and their intracellular localization. By contrast, all other αS species, including LF, caused negligible calcein release at this incubation time (Fig. 3a).

Accordingly, a similar pattern of cytosolic $Ca^{2+}$ dyshomeostasis was observed for the different αS species in cortical neurons and SH-SY5Y cells (Fig. 3b), as previously reported[13,30,31]. In particular, following 15 min treatment at an αS concentration of 0.3 µM, OB* and, to a lesser extent, SF and even lesser for LF, triggered a significant influx of $Ca^{2+}$ into cortical neurons and SH-SY5Y cells ($551 \pm 31\%$, $356 \pm 50\%$, $261 \pm 16$ and $380 \pm 16$, $230 \pm 27$ and $124 \pm 11$ for the three species and two cell types, respectively), whereas all other αS species caused a negligible $Ca^{2+}$ influx (Fig. 3b). No fluorescent signal was apparent when SH-SY5Y cells were cultured in a $Ca^{2+}$-free CM (Figure S6a,b). Lower effects were observed at lower and higher protein concentrations with a clear bell-shaped dose response (Figure S6a,b).

Real-time intracellular calcium measurements in SH-SY5Y living cells showed a steady basal fluorescence in the absence of extracellular aggregates and a rapid increase in the intracellular $Ca^{2+}$ levels following the addition of OB* and, with slower kinetics, of SF up to 30 min (Figure S6c). Notably, we also observed a faster increase of intracellular $Ca^{2+}$ for OB* than SF and LF treatment at longer time periods in SH-SY5Y cells (Fig. 3c).

We further monitored whether the ability of αS aggregates to cause ionic dyshomeostasis resulted in the activation of caspase-3, a well-recognized apoptotic marker[32]. To this aim, we used a relevant PD model such as human iPSC-derived dopaminergic neurons expressing TH, a typical dopaminergic neuron marker (~75% of total cells), as well as the microtubule-associated protein 2 (MAP-2), a typical mature neuron marker, after 14–18 days of culture (Fig. 3d). A significant caspase-3 activation ($444 \pm 13\%$) was found in iPSC-derived dopaminergic neurons following treatment for 24 h with 0.3 µM OB* with respect to untreated cells (Fig. 3e, f). Both SF and LF caused a caspase-3 activation of $337 \pm 31\%$ and $265 \pm 9\%$, respectively, in the same neurons after 24 h of treatment (Fig. 3e, f). Accordingly, similar results were observed in SH-SY5Y cells treated with OB*/SF/LF, whereas no apoptosis was apparent in cells treated with 0.3 µM OA* or M (Fig. 3f).

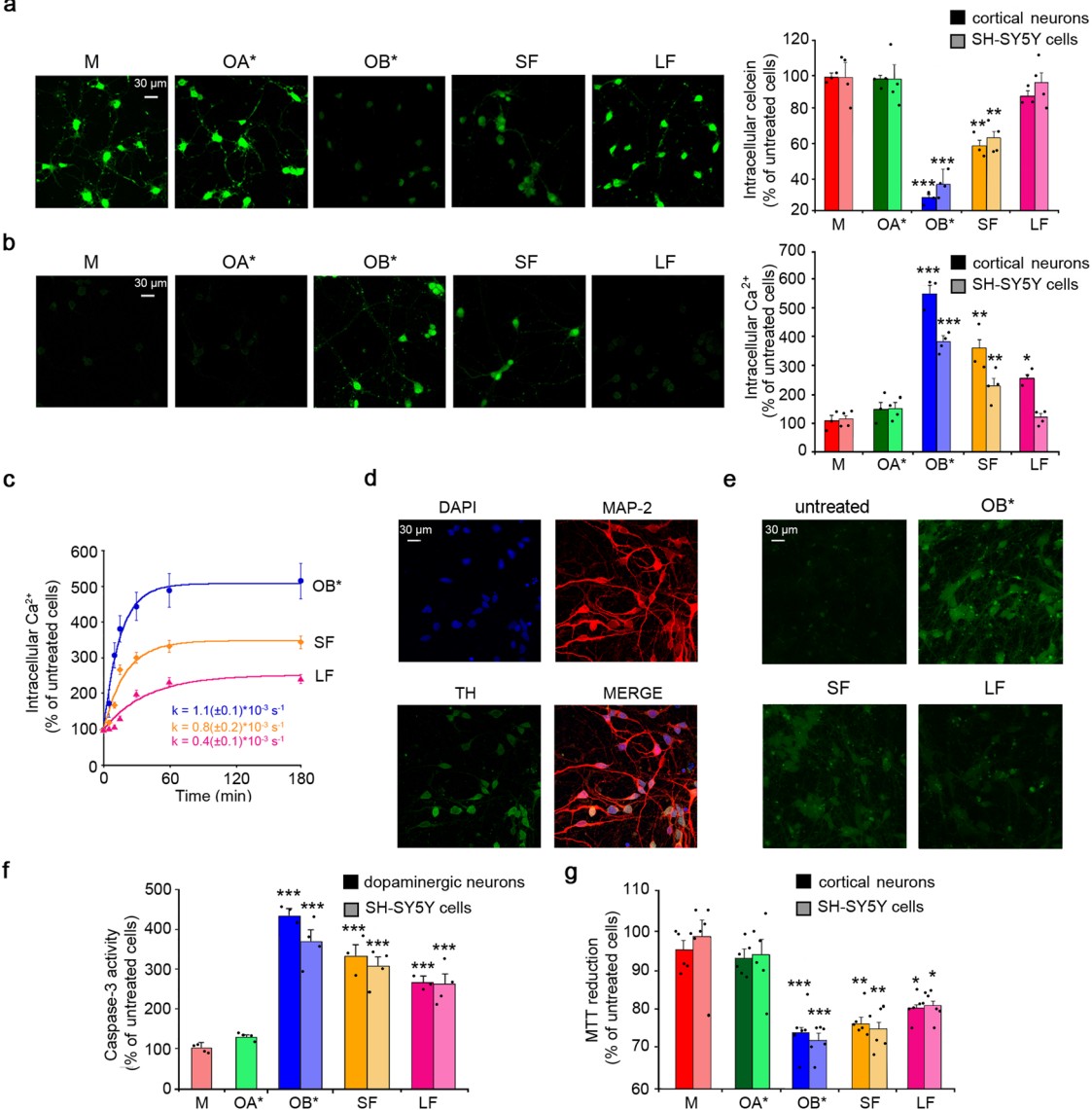

**Fig. 3 αS fibrils gradually destabilize membrane integrity resulting in neuronal dysfunction. a** Representative confocal microscope images showing primary rat cortical neurons loaded with the calcein-AM probe for 10 min and then treated for 1 h with the indicated 0.3 μM αS species. Semi-quantitative analyses of the calcein-derived fluorescence signal in primary rat cortical neurons and SH-SY5Y cells. **b** Representative confocal microscope images showing the $Ca^{2+}$-derived fluorescence in primary rat cortical neurons treated for 15 min with the indicated 0.3 μM αS species and then loaded with the Fluo-4 AM probe. Semi-quantitative analysis of the intracellular $Ca^{2+}$-derived fluorescence in primary rat cortical neurons and SH-SY5Y cells. **c** Time-course analysis of the intracellular $Ca^{2+}$-derived fluorescence in SH-SY5Y cells treated for the lengths of time indicated with OB*/SF/LF at 0.3 μM. **d** Representative confocal microscope images of human iPSC-derived dopaminergic neurons expressing MAP-2 (ab32454, Abcam) and TH (sc-25269, Santa Cruz Biotechnology) markers at 14–18 days of maturation (three independent experiments with one internal replicate). Approximately 75% of the cells are TH positive (estimated by immunostaining). Nuclei were stained with DAPI. **e** Representative confocal microscope images showing caspase-3-derived fluorescence in human iPSC-derived dopaminergic neurons treated for 24 h with the indicated αS species at 0.3 μM. **f** Semi-quantitative analysis of the caspase-3-derived fluorescence in human iPSC-derived dopaminergic neurons and SH-SY5Y cells treated for 24 h with the indicated αS species at 0.3 μM. **f** MTT reduction in primary rat cortical neurons and SH-SY5Y cells treated for 24 h with the indicated 0.3 μM αS species. In all panels data are expressed as the percentage of the value for untreated cells. Experimental errors are S.E.M. ($n = 3$ with two internal replicates and $n = 4$ with three internal replicates for cortical neurons and SH-SY5Y cells, respectively, in panels (**a**), (**b**); $n = 4$ in panel **c** with one internal replicate; $n = 3$ with two internal replicates and $n = 4$ with three internal replicates for iPSC-derived dopaminergic neurons and SH-SY5Y cells, respectively, in panel **f**; $n = 6$ with three internal replicates in panel (**g**)). Samples were analyzed by one-way ANOVA followed by Bonferroni's multiple comparison test relative to untreated cells (in panels **a**, **b**, **f**, and **g**, *$P < 0.05$, **$P < 0.01$, ***$P < 0.001$). A total of 200–250 cells (**a–f**) and 150,000–200,000 cells (**g**) were analyzed per condition.

A consistent trend was observed for the mitochondrial dysfunction, as revealed by the 3-(4,5-dimethylthiazol-2-yl)-2,5-diphenyltetrazolium bromide (MTT) reduction inhibition assay[33] carried out on both primary rat cortical neurons and SH-SY5Y cells (Fig. 3g). Since αS is physiologically N-terminally acetylated in vivo[34], representative experiments were repeated with OB*

formed from the N-acetylated αS. It has been recently reported that no significant differences were observed in the structural properties of αS species following N-terminal acetylation[8]. Consistent results were obtained in the present study, using calcein release and MTT reduction as cell viability readouts (Figure S7a,b). Similar results were also obtained with the ultra-

clean samples, where the recombinant protein was pretreated to remove potential lipopolysaccharides, probing ROS detection and MTT reduction (Figure S7c,d).

Taken together our results indicate that αS SF have lower ability than OB* to cause fast membrane permeabilization, rapid intracellular $Ca^{2+}$ influx and ROS generation in primary neurons and SH-SY5Y cells, with LF being almost inert; however, SF have effects closer to OB* and even LF show some effect when downstream cellular dysfunction, such as metabolic capacity and apoptotic response, was probed at longer time frames.

**αS fibrils gradually release oligomers in vitro**. We took advantage of the conformation-sensitive A11 antibody to investigate whether the release of toxic oligomers from αS fibrils is responsible for the observed toxicity of SF. The polyclonal A11 antibody, originally raised against prefibrillar Aβ oligomers, has been shown to recognize toxic oligomers from various proteins, but not monomers or fibrils[35]. The A11 antibody showed a high affinity for OB*, whereas the absence of any A11-positive cross-reaction of SF/LF rules out the possibility that a significant fraction of OB* species was present in the fresh fibrillar samples at micromolar concentrations (Fig. 4a). Nor did it react with control nontoxic OA*.

We monitored the ability of αS fibrils to release small oligomers over time in the absence of cells. Confocal images of SF samples, incubated in the CM without cells in wells containing a glass coverslip for 0, 1, 3, and 24 h at 37 °C, resolved by the conformation-insensitive mouse monoclonal 211 antibody showed similar populations of αS-derived fluorescence dots (Fig. 4b). By contrast, αS-derived fluorescence dots increased in number with incubation time at 37 °C in SF using the conformation-sensitive A11 antibody, suggesting a slow and progressive increase in A11 immunoreactivity (Fig. 4b). A slower time-dependent increase in A11-positive signal was also obtained with LF following incubation up to 24 h (Figure S8a). Very similar small dots were apparent in confocal images of OB* sample using both conformation-insensitive 211 and conformation-sensitive A11 antibodies following incubation for 0 and 24 h at 37 °C (Fig. 4b and Figure S8a). No fluorescent signal was evident with primary and secondary antibodies in the absence of αS aggregates, ruling out the ability of the antibodies to react with the slides in the absence of αS species (Figure S8a).

To confirm that the A11 reactivity in aged SF/LF originates from the release of small oligomers rather than from a conformational change in SF exposing the A11 epitope, we also analyzed SF upon incubation for 24 h at 37 °C using dynamic light scattering (DLS) (Figure S8b). The SF sample at 0 h showed a large peak at ~600 nm, but a small peak at ~20 nm became increasingly apparent with time (Figure S8b). The size distribution in the small peak was consistent with that detected in OB* sample incubated up to 24 h at 37 °C (Figure S8b). This indicates that αS fibrils can release A11-positive oligomeric species following incubation at 37 °C in vitro.

**αS fibrils gradually release oligomers that are ultimately responsible for their toxicity**. Then, we investigated whether αS SF can release small oligomers following their interaction with cellular membranes. We probed the nature of the protein species able to penetrate into human iPSC-derived dopaminergic neurons using the A11 antibody in immunofluorescence experiments (Fig. 4c). Confocal images showed a remarkable A11-positive signal in neurons treated for 24 h with OB*, SF, and LF (Fig. 4c) with respect to untreated cells (Figure S8c), indicating the presence of αS oligomeric species similar to OB* inside the cells upon treatment with SF/LF. When SH-SY5Y cells were exposed

to OB* for 3 h, a dramatic increase (474 ± 26%) in A11-positive intracellular species at median planes was observed with respect to untreated cells (Fig. 4d, e and Figure S8c). In particular, ~62% of the OB* sample was found to penetrate the cells (defined as the ratio between the intracellular fluorescence at the median planes and the total fluorescence at all planes), in good agreement with the fraction of intracellular protein obtained by αS conformation-insensitive antibody (Ab52168) (Fig. 2a). Cells exposed to SF for 3 h and to LF for 6 h also exhibited a significant increase (258 ± 6% and 199 ± 13%, respectively) in the intracellular A11-positive signal (Fig. 4d, e), indicating a progressive penetration of αS oligomeric species similar to OB*. Indeed, the majority of the A11-derived fluorescence signal was located in the interior of the cells (87 ± 7% and 97 ± 6% for SF and LF, respectively), suggesting that the A11-positive oligomers had been able to cross the membrane.

To investigate whether the intracellular A11-positive signal could arise from the conversion of endogenous αS into A11-positive species, we first quantified the intracellular and total fluorescences at different OB* doses (Figure S9a-c). A bell-shaped protein concentration dependence was found for both, with the amount of the A11-positive species inside the cells found to be directly proportional to the mass concentration of those species interacting with the plasma membrane (Figure S9a-c), confirming the results obtained with a conformation-insensitive antibody (Figure S4a-e). Second, we quantified the intracellular and total A11-derived fluorescence following 15 min and 24 h of OB* treatment (Figure S9d). Neither the intracellular nor the total A11-derived fluorescence at 24 h exceeded the total A11-derived fluorescence observed early at 15 min, suggesting that the endogenous αS was not recruited into A11-positive species.

When we monitored the intracellular A11-positive signal as a function of time, we found exponential kinetics for OB*, but sigmoidal kinetics for both SF and LF (Fig. 4e). The results obtained with the A11 antibody were also confirmed with another conformational-sensitive antibody such as Syn33, that has been previously shown to specifically recognize αS oligomers[36]. Here, the Syn33 antibody showed a high affinity for OB* and a very low cross-reaction with SF/LF in the dot-blot assay (Figure S8d). When SH-SY5Y cells were exposed to OB* for 3 h, a dramatic increase (407 ± 24%) in Syn33-positive intracellular species at median planes was observed with respect to untreated cells (Figure S8e,f), in good agreement with the immunostaining data obtained by A11 antibody (Fig. 4c-e). Cells exposed to SF for 3 h and to LF for 6 h also exhibited a significant increase (177 ± 22% and 173 ± 13%, respectively) in the intracellular Syn33-positive signal (Figure S8e,f). These results are in agreement with the idea that a lag time is required for oligomer release from SF/LF before cell penetration.

To have equivalent molar concentrations of fibril particles (numbers/volume) rather than monomer equivalents, we have estimated the same fibril number concentrations for SF/LF (corresponding to a higher mass concentration for LF, see "Methods"). We have thus chosen two different mass concentrations where the fibril particle concentrations are similar (0.03 μM SF and 0.3 μM LF, both monomer equivalents), finding similar toxic effects using ROS and $Ca^{2+}$ probes (Figure S8g).

As αS toxicity depends on the time of cellular exposure to the deleterious αS species, we monitored MTT reduction following incubation of SH-SY5Y cells for different lengths of time with 0.3 μM OB*/SF/LF. We observed a similar order of rates for the three species, with the OB* being the most rapid and the LF being the slowest in causing a decrease of MTT reduction (Fig. 4f). Moreover, a plot of the degree of MTT reduction against the intracellular A11-positive αS levels, using the data obtained at all time points and for OB*/SF/LF, revealed a very significant

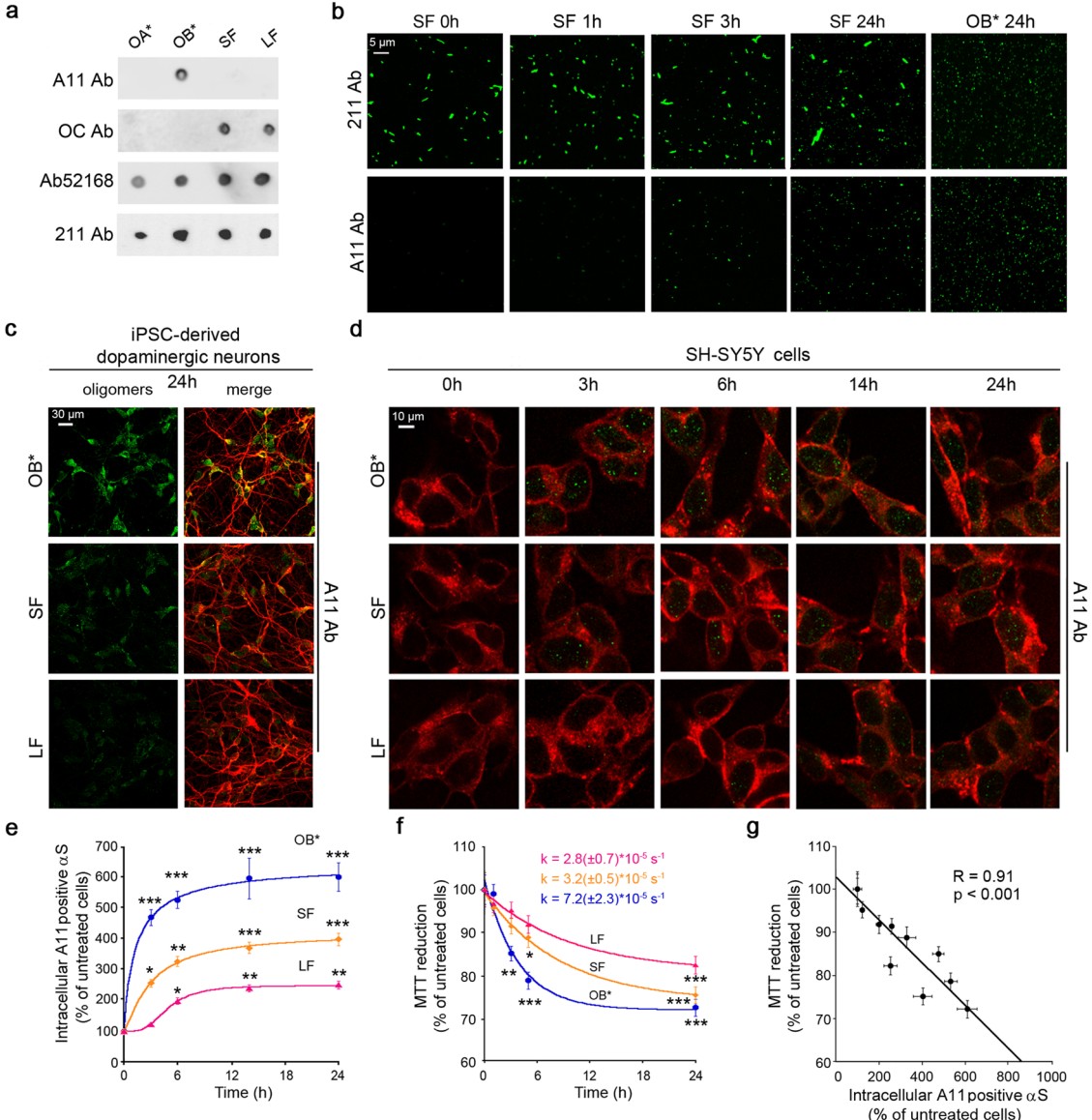

**Fig. 4 αS fibrils gradually release oligomers that are ultimately responsible for their toxicity. a** Dot-blot analysis of αS species probed with conformational specific antibody A11 (AHB0052, Thermo Fisher Scientific), OC (AB2286, Sigma-Aldrich), conformation-insensitive polyclonal anti-αS antibody (ab52168, Abcam) and conformation-insensitive monoclonal 211 antibody specific for human αS (sc12767, Santa Cruz Biotechnology). **b** Representative confocal microscope images showing SF (at 0.3 μM) incubated in CM without cells in wells containing a glass coverslip for 0–24 h at 37 °C. Representative images of OB* incubated for 24 h were also shown as positive control. The green-fluorescent signals derive from the staining with mouse monoclonal 211 anti-αS antibodies and rabbit anti-oligomer A11 polyclonal antibodies, in the first and second rows, respectively, and then Alexa-Fluor 514-conjugated anti-mouse or anti-rabbit secondary antibodies (three independent experiments with one internal replicate). **c** Representative confocal scanning microscope images showing human iPSC-derived dopaminergic neurons treated for 24 h with OB*/SF/LF at 0.3 μM. Red and green fluorescence indicates mouse anti-MAP-2 antibodies (ab11267, Abcam) and the A11-positive prefibrillar oligomers, respectively. **d** Representative confocal scanning microscope images showing the median sections of SH-SY5Y cells treated for the lengths of time indicated with OB*/SF/LF at 0.3 μM. Red and green fluorescence indicates the cell membranes labeled with WGA and the A11-positive prefibrillar oligomers, respectively (three independent experiments with four internal replicates). **e** Kinetic plots reporting A11-intracellular fluorescence following the addition of 0.3 μM of the indicated αS species to SH-SY5Y cells. The continuous lines through the data represent the best fits to exponential and sigmoidal functions (see "Methods"), for OB*, SF, and LF, respectively. **f** Kinetic plots reporting the MTT reduction versus time elapsed following addition of 0.3 μM of the indicated αS species to SH-SY5Y cells. **g** Dependence of MTT reduction on the penetration of A11-positive αS in SH-SY5Y cells treated with αS species. MTT reduction values reported in (**f**) plotted against the αS-derived intracellular fluorescence values reported in (**e**) of cells treated with OB*/SF/LF at the corresponding times. Experimental errors are S.E.M. ($n = 3$ with four internal replicates in panel (**e**); $n = 4$ with three internal replicates in panel (**f**); $n = 3$ with four internal replicates and $n = 4$ with three internal replicates for MTT reduction and intracellular A11-positive aS, respectively, in panel (**g**). Samples were analyzed by one-way ANOVA followed by Bonferroni's multiple comparison test relative to untreated cells (in panels **e** and **f**, *$P < 0.05$, **$P < 0.01$, ***$P < 0.001$; in panel **g**, $P < 0.001$). A total of 200–250 cells (**e**) and 150,000–200,000 cells (**f**) were analyzed per condition.

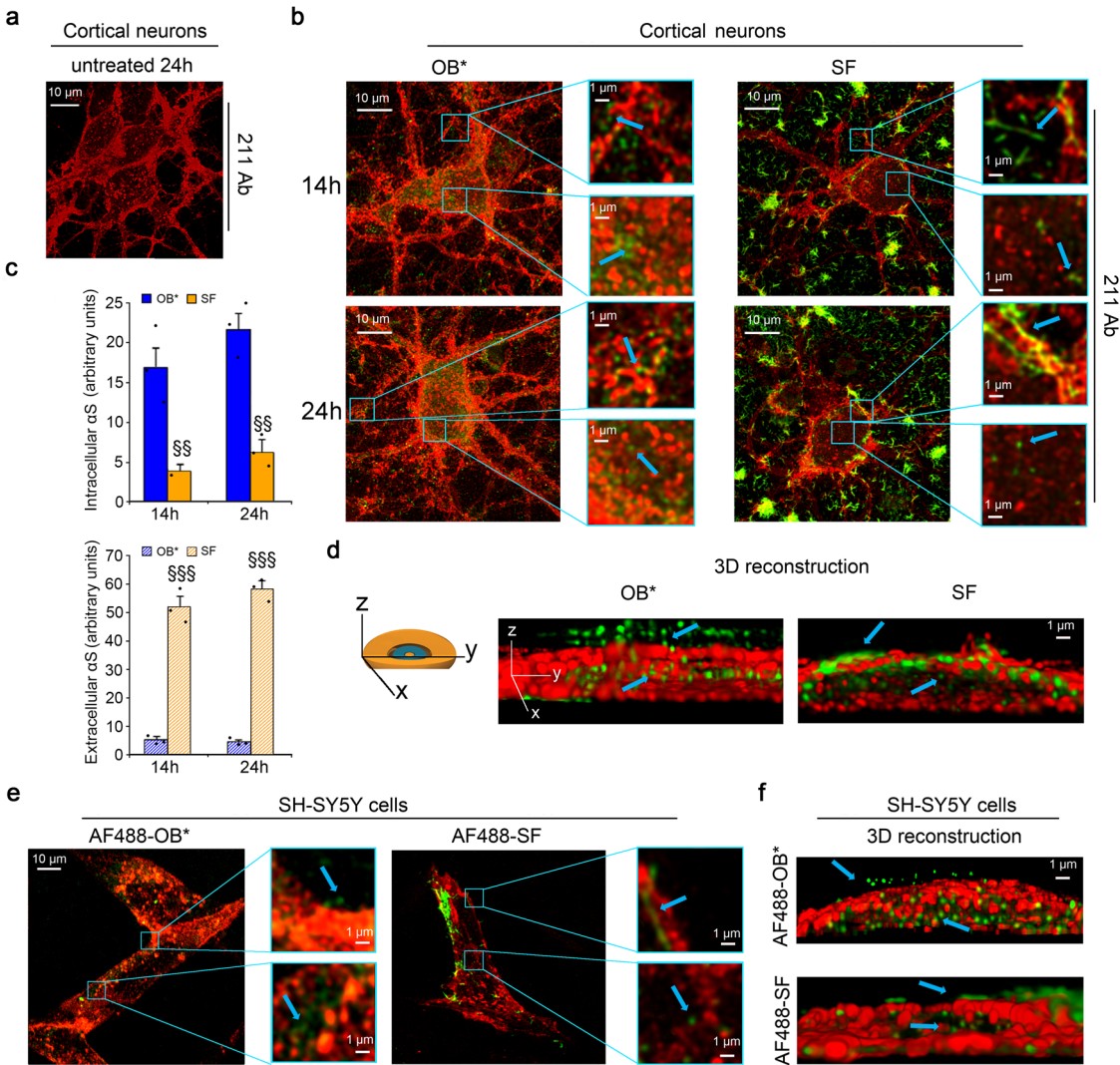

**Fig. 5 Visualization of αS fibrils outside and oligomers inside cells at high resolution. a, b** Representative STED images of primary rat cortical neurons that were untreated (**a**) or treated (**b**) with OB* (left) and SF (right) for 14 and 24 h. Red and green fluorescence indicates the cell membranes and the αS species revealed by WGA and the conformation-insensitive and human αS-specific 211 antibodies (sc12767, Santa Cruz Biotechnology), respectively. Higher magnifications of the αS species are shown in the boxed areas (three independent experiments with two internal replicates). **c** Semi-quantitative analysis of the intracellular and extracellular 211-derived fluorescence data referring to panel (**b**). Experimental errors are S.E.M. ($n = 3$ with two internal replicates). Samples were analyzed by one-way ANOVA followed by Bonferroni's multiple comparison test relative to cells treated with OB* for 14 and 24 h (in panel **c**, $^{§§}P < 0.01$, $^{§§§}P < 0.001$). **d** 3D reconstruction of the z-stack analysis (5-µm-thick slices) of the specimens shown in panel (**b**). A primary neuron was virtually dissected on the *zy* plane to show more clearly the extracellular (top) and intracellular (middle) αS species. A total of 40–60 cells were analyzed per condition (three independent experiments with four internal replicates). **e** Representative STED images of SH-SY5Y cells that were treated with AF488-OB* (left) and AF488-SF (right) for 24 h. Red and green fluorescence indicates the cell membranes labeled with WGA and αS labeled with AF488 dye, respectively. Higher magnifications of the αS species are shown in the boxed areas (three independent experiments with one internal replicate). **f** 3D reconstruction of the z-stack analysis (5-µm-thick slices) of the specimens shown in panel (**e**). Other details as in panel d (three independent experiments with one internal replicate). In all panels blue arrows indicate either fibrillar or oligomeric αS species.

negative correlation (R = 0.91, $p < 0.001$, Fig. 4g), indicating that the amount and kinetics of released oligomers match the observed levels and kinetics of toxicity in cellular systems.

**Visualization of αS fibrils outside and oligomers inside cells at high resolution.** To evaluate further the nature of the αS species that were found inside the cells, we used the super-resolution STED microscopy[37] on rat primary cortical neurons and the mouse monoclonal 211 anti-αS antibody (Fig. 5a–d). This conformation-insensitive antibody was raised against residues 121–125 of human αS (sequence DNEAY), so is unable to interact

with endogenous rat αS (sequence SSEAY). Accordingly, rat cortical neurons treated with the antibody in the absence of exogenous human αS did not reveal any green fluorescence arising from endogenous αS (Fig. 5a). By contrast, neurons exposed to human αS-derived OB* for 14 and 24 h exhibited green-fluorescent punctae, which appeared to be small and globular at the very high magnifications allowed by STED microscopy (Fig. 5b, arrows in the left image magnification). Neurons treated with SF for 14 and 24 h, however, showed αS aggregates outside the cells or attached to the membrane that, at high magnification, appeared fibrillar in morphology with a length distribution of the same order of magnitude as that

determined by AFM (Fig. 5b, arrow in the top right image magnification). The same images also showed a number of small and globular intracellular αS species, with a morphology that resembled the intracellular OB* (Fig. 5b, arrow in the bottom right image magnification). These can only include exogenous human αS given the specificity of the 211 antibody. The 211-derived green-fluorescent signals of cortical neurons treated with OB* and SF for 14 and 24 h were consistent with the results obtained with Ab52168 and A11 antibodies (Fig. 5c), although the reliability of STED quantification is limited by the selective photobleaching of fluorophores coupled with super-resolution microscopy.

When z-stack analysis was used to perform the 3D reconstruction of the primary neurons treated with OB*, small oligomeric species attached on the outer leaflet of the plasma membrane (small green punctae on the top) and penetrated into the cytosol (small green punctae in the middle) were evident (arrows in Fig. 5d). By contrast, neurons treated with SF showed fibrillar species attached to the extracellular side of the cell membrane along with intracellular globular species similar to OB* (arrows in Fig. 5d). STED imaging was repeated on human SH-SY5Y cells treated with OB* and SF labeled with the fluorescent AF488 dye, clearly showing small oligomers with globular morphology on the membrane surface and inside the cells and only inside the cells, respectively (Fig. 5e, f). We performed this experiment with a low ratio of AF488 labeled OB* and SF (generated by mixing 90 and 10% of unlabeled and labeled αS monomers), thus ensuring the absence of significant modifications to the properties of OB* and SF as previously reported[6,8].

These findings reveal that the αS species detected in neurons and SH-SY5Y cells by confocal microscopy and STED using different probes arise from exogenous human αS derived from SF/LF, rather than endogenous rat αS converted in oligomeric form by a seeding process. Moreover, to assess whether the toxicity induced by fibrils could originate from a released M that could progressively convert into oligomers by a process of secondary nucleation[27], we exposed the SH-SY5Y cells to SF (0.03 μM) for 15 or 180 min in the absence or presence of a 10-fold excess of M (0.3 μM). We have chosen a time of 180 min, as the toxicity observed with cells exposed to SF/LF was already evident within 180 min (Fig. 3c). The results did not show any significant difference in the degree of $Ca^{2+}$ influx between the different conditions (Figure S10), indicating that the oligomers observed are directly generated by the disaggregation or fragmentation of fibrils under the conditions of study.

**Inhibition of oligomer release from fibrils prevents their toxicity.** To provide further evidence that the fibrils release toxic oligomers and that these are the major species responsible for the αS toxicity, we performed an additional set of competitive experiments. SH-SY5Y cells were exposed to OB*/SF/LF (0.3 μM) and a 2.5-fold excess of the A11 antibody was added to the CM for 24 h. Without antibodies, cells treated for 24 h with OB* and, to a lesser extent SF/LF, showed intracellular small green dots, as revealed by the Ab52168 antibody (Fig. 6a, b). Addition of A11 antibody to the CM containing αS species significantly reduced the OB* penetration by ~68% (Fig. 6a, b) and abolished completely the OB* cytotoxicity (Fig. 6c). The reduction of small green dots in the cytosol of A11-treated cells exposed to SF and LF (by ~80% and ~77%, respectively) also resulted in a complete recovery of cell viability (Fig. 6a–c). This data suggest that A11 antibody binds oligomeric species that are released from SF/LF, thus preventing their cellular uptake and their downstream effects. We repeated the experiments using the conformation-sensitive OC antibody, raised against fibrillar species[38], found

here to react with SF/LF, but not OB* (Fig. 4a). The addition of OC antibody to the CM containing SF/LF reduced significantly the intracellular αS-specific antibody signal (by ~75% and ~63%, respectively) and their cytotoxicity, without affecting the intracellular fluorescence (reduced only by ~7%) and the toxicity (reduced only by ~3%) when added to the CM containing OB* (Fig. 6a–c).

Moreover, a plot of the degree of MTT reduction against the intracellular αS levels using the data obtained for OB*/SF/LF with or without A11/OC antibodies, revealed a very significant negative correlation (R = 0.88, p < 0.001) (Fig. 6d), confirming that the most inherently toxic αS species are β-sheet containing oligomers, which can either be formed during the self-assembly of the protein into amyloid fibrils or released from fibrillar species once they are formed.

## Discussion

In this work, NMR and CD data show that the β-sheet core of the αS fibrils is unable to establish persistent interactions with either the surface or the internal regions of the lipid bilayers, with the interactions restricted to the binding of the disordered N-terminal region of αS onto the surfaces of the lipid bilayers. Accordingly, following their addition to the CM, the SF and LF forms of αS appear to be largely localized at the surface of the plasma membrane on the timescale of the experiments studied here, at which toxicity is observed. The total quantity of SF/LF bound to the cell membrane was not found to correlate with the degree of cell dysfunction or αS penetration. Therefore, the association between SF or LF and cellular membranes is not sufficient to account for the toxicity of these species. On the contrary, most of the prefibrillar OB* species that interact with the neuronal membrane appear to penetrate into the cytosol, and the amount of intracellular OB* was found to be directly proportional to the mass concentration of the species that interact with the plasma membrane and the resulting toxicity.

The biological analyses revealed that prefibrillar OB* species induce a rapid dysregulation of calcium and redox status homeostasis, in agreement with our previous results[6,8,12,13,39]. Although the fate of αS aggregates once they enter the cells needs further investigation, the influx of OB* through the membrane bilayer is partially mediated by endocytosis. The uptake of OB* was found to result in further membrane disruption (as indicated by calcein release), mitochondrial dysfunction, and apoptosis. Importantly, the same cascade of events was observed for the fibrillar samples, although with remarkably different kinetics. Our results, therefore, indicate that αS aggregates with a β-sheet core and solvent-exposed hydrophobic surfaces appear to be able to induce cellular damage, although the time scales and levels of observed toxicity depend on the nature (oligomeric vs fibrillar) and size of the aggregates.

Thus, the question remains as to how fibrillar species exert their toxicity. According to our results, the toxic effects of the fibrils appear to correlate directly with the quantity of αS that penetrate into the cells after interacting with the cellular membranes and cause calcium uptake, ROS formation, membrane permeabilization to calcein, caspase-3 activation, and mitochondrial dysfunction. The dissociation of αS fibrils into soluble αS species, likely to include a low proportion of monomers in addition to oligomers, have also been observed in recent studies to occur under conditions close to physiological[6,40]. Using confocal microscopy and the A11 and Syn33 antibodies, which are specific for 'soluble prefibrillar oligomers'[35,36], we found that the species that can penetrate into the cells were A11- and Syn33-positive in all cases, not only when the cells were treated with OB*, but also with SF and LF species that are A11- and

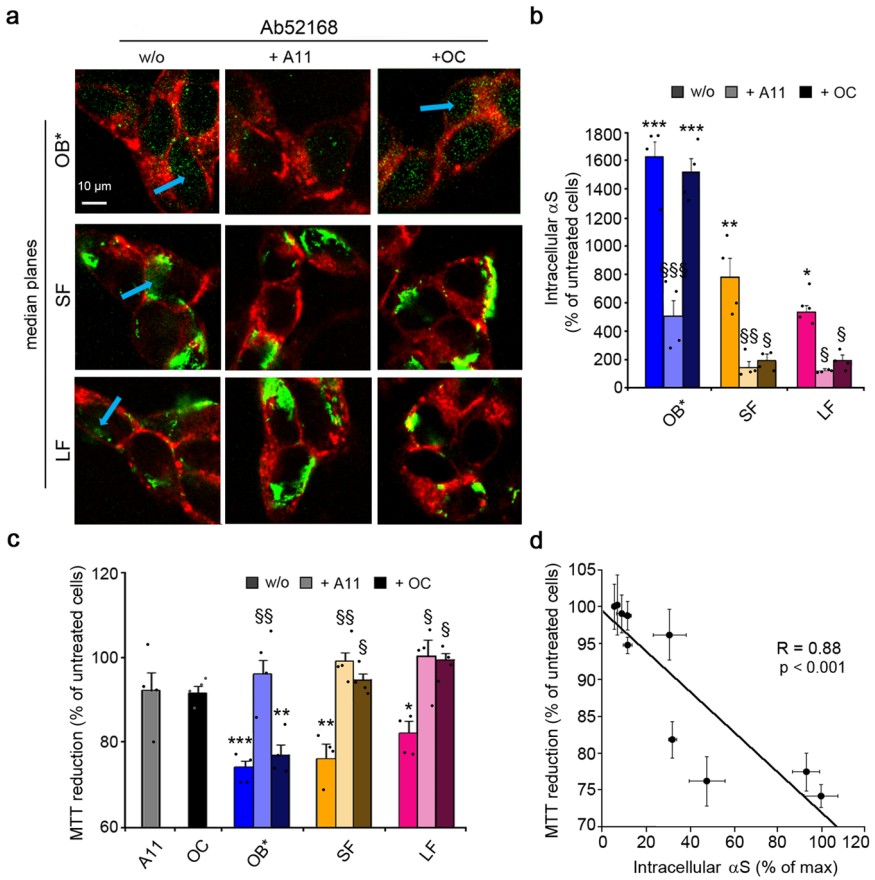

**Fig. 6 Inhibition of oligomer release from fibrils prevents their toxicity. a** Representative confocal scanning microscope images showing the median sections of SH-SY5Y cells treated for 24 h with the indicated αS species at 0.3 µM, in the absence or presence of A11 (AHB0052, Thermo Fisher Scientific) and OC (AB2286, Sigma-Aldrich) antibodies at 1:2.5 molar ratio. Red and green fluorescence indicates the cell membranes and the αS species revealed with WGA and polyclonal anti-αS antibodies (ab52168, Abcam), respectively. The arrows in the images show the intracellular green-fluorescent punctae. **b** Semi-quantitative analysis of the green fluorescence signal referring to panel (**a**) and derived from intracellular αS species expressed as the percentage of untreated cells. **c** MTT reduction in SH-SY5Y cells treated for 24 h with the indicated αS species at 0.3 µM in the absence or presence of A11 and OC antibodies (1:2.5 molar ratio). **d** Dependence of MTT reduction (values reported in panel **c**) on the αS-derived fluorescence values in cells treated with OB*/SF/LF (values from panel **b**) in the absence or presence of A11 and OC antibodies (1:2.5 molar ratio). In all panels, experimental errors are S.E.M. (n = 4 with three internal replicates). Samples were analyzed by one-way ANOVA followed by Bonferroni's multiple comparison test relative to untreated cells (in panels **b** and **c**, *$P < 0.05$, **$P < 0.01$, ***$P < 0.001$) and cells treated with the αS species (in panels **b** and **c**, §$P < 0.05$, §§$P < 0.01$, §§§$P < 0.001$). In panel **d**, $P < 0.001$. A total of 200–250 cells (**a**, **b**), and 150,000–200,000 cells (**c**) were analyzed per condition.

Syn33-negative per se. In a cell-free context, SF/LF were found to slowly release A11-positive species and DLS indicated a slow release of small-sized species. The appearance of A11-positive aggregates from the fibrillar species was relatively slow, with rates and overall kinetic traces that closely match the time course of toxicity for SF/LF. This also explains why fibrils caused a minor and slower calcein release from SUVs. Finally, using STED super-resolution microscopy in rat primary cortical neurons coupled with 3D analysis and the monoclonal 211 antibody that recognizes only the exogenous human αS, we provide further evidence for the oligomeric, rather than fibrillar, morphology of the αS species released from exogenous αS fibrils upon interaction with the cell membrane and later entering the cytosol. When z-stack analysis was used to perform the 3D reconstruction, the release of oligomeric species from SF was also evident inside the cells exposed to the fluorescently labeled AF488-SF in the absence of any antibody.

Our finding that LF are less toxic than SF can be attributed to their lower proportion of fibrillar ends at the same mass concentration. As oligomer release occurs from fibril ends, this feature will result in a slower release of oligomers. Moreover, LF

showed a lower diffusion capacity in the CM, and a reduced ability to interact with the cells, than SF. These conclusions are in line with previous reports obtained with the amyloid-beta (Aβ) peptide associated with Alzheimer's disease that have revealed the lipid-mediated depolymerization of nontoxic fibrils of Aβ into toxic A11-positive oligomers, which were also shown to resemble the oligomers formed de novo during fibril assembly[41]. They are also in agreement with the general proposition that any fibrillar species that accumulate in tissue can represent a source of soluble toxic oligomers[6,42], and with the halos of soluble oligomers observed to surround amyloid plaques of Aβ in mouse brains[43].

In a recent study, we have found that SF species analyzed in the present paper are able to induce PD-like pathology by a spreading process when injected into the brains of healthy mice, whereas the OB* species were only able to induce local toxicity in the region of brain injection, without signs of seeding and spreading, as it was expected, given their kinetically trapped nature[24]. It is possible, however, that the OB-like species that we have observed to be released from fibrils in the present study, in addition to contributing to the induction of toxicity, could also enhance the generation of new aggregates, as they might have elongation

capabilities similar to those of their fibrillar precursors, unlike OB* species[4].

In conclusion, our results suggest that αS fibrils, particularly the small ones, have a doubly deleterious effect. The first has been explored previously and results from a transfer from neuron-to-neuron contributing to the slow but progressive diffusion of Lewy body pathology in different brain areas. Indeed, some αS fibril polymorphs have been observed to be internalized by cells[22,23] and possibly contribute to pathology spreading[17–21,24]. The second effect stems from the release of prefibrillar oligomeric species causing an immediate dysfunction of the neurons in the vicinity of these species. Such oligomeric species could also contribute to pathogenesis via neuron-to-neuron spreading by their direct cell-to-cell transfer or by generating new fibrils, following their neuronal uptake.

Thus, molecules able to prevent the formation of toxic oligomers of αS during the aggregation process, such as aminosterols[44–46], or to interact directly with such species, for example the A11 antibody, or to prevent their release from the fibrillar species, such as the OC antibody, are able to suppress αS toxic behavior. Indeed, these three mechanisms have been shown to be used by different molecular chaperones to protect cells from amyloid aggregation, some inhibiting oligomer formation, other binding to and neutralizing the effects of preformed oligomers, and other binding preferentially to fibril ends[47–51]. These features are of considerable interest in the design of therapeutic approaches targeting α-synucleinopathies and other systems in which protein aggregation is linked to neurodegeneration.

## Methods

**Purification of αS monomers and formation of αS OA* and OB*, SF and LF**. The recombinant human αS was over-expressed in *E. coli* BL21 cells (Agilent, UK) and purified as a monomeric fraction[4]. OA* were generated by incubating ca. 200 µM of αS in PBS pH 7.4 with ten molar equivalents of (-)-epigallocatechin-3-gallate (EGCG) (Merck, Darmstadt, Germany) for 48 h at 37 °C. After the incubation, excess of compound and monomeric protein was removed by six consecutive cycles of filtration through 100 kDa centrifuge filters (Merck). For the isolation of OB*, purified αS was dialyzed against miliQ water and lyophilized for 48 h in aliquots of 6 mg. The aliquots were resuspended in 500 µL of PBS pH 7.4 to a final concentration of 800 µM, filtered through a 0.22 µm filters and incubated at 37 °C without agitation for 20–24 h. Resulting fibrils formed during the incubation were removed by ultracentrifugation at 288,000 × g. Monomeric protein and small oligomers were removed by four consecutive cycles of filtration through 100 kDa centrifuge filters (Merck)[4,8]. Final oligomer concentration was determined measuring the absorbance at 280 nm and using the extinction coefficient 5960 M⁻¹ cm⁻¹ for the OB*, or by BCA analysis for OA* oligomers (in this later case, the yield of production was consistently ca. 100%). Both oligomeric samples were kept at room temperature and were used within 3 days after their production[4,8]. Long fibrillar samples (f0) were prepared by incubating monomeric αS at 70 µM (1 mg/ml) in PBS buffer pH 7.4 (0.1 M ionic strength) containing 0.01% NaN₃ at 37 °C, under constant agitation (New Brunswick Scientific Innova 43, 200 rpm) for 4–6 days. After this time, each sample was centrifuged (15 min at 16,100 × g) and the fibrillar pellet washed twice with PBS before being resuspended into the appropriate volume of PBS. As clumping of fibrils was observed[52] a second generation of long fibrils (f1) was prepared by incubating 100 µM monomeric αS with 10 µM of f0 sonicated for 1 min (Bandelin, Sonopuls HD 2070, cycles of 0.3 s of active sonication followed by 0.7 s of passive interval, 10% maximum power) in 500 µL PBS at 37 °C under quiescent conditions for 13–15 h and then centrifuged as described above for f0. Samples of short f1 fibrils were generated by sonicating long f1 fibrils for 20 s using the same apparatus and settings. The concentration of fibrils was estimated by measuring the absorbance at 275 nm using ε₂₇₅ = 5600 M⁻¹ cm⁻¹ after disaggregating an aliquot by the addition of guanidinium chloride to a final concentration of 4 M.

N-terminally acetylated αS was obtained by coexpression of a plasmid carrying the components of the NatB complex (Addgene) with the pT7-7 plasmid encoding for human WT αS. The purification protocol of N-terminally acetylated protein was identical to the non-modified variant and the yield of acetylation was assessed by mass spectrometry and resulted to be essentially complete.

In a set of experiments, LPS (lipopolysaccharide) ultra-clean, also referred to as LPS-free, OB* samples were used to study the possible effect of LPS residual concentrations present in the OB* samples generated from the recombinant protein. For this, a set of commercial kits to remove the LPS endotoxin and quantify its content in the different protein samples (Thermo Fisher Scientific, Waltham, MA, USA) was used.

Fluorescently labeled αS molecules carrying the AF488 dye (Invitrogen, Carlsbad, CA, USA) were obtained by using the N122C mutational variant[6]. The labeled protein was then purified from the excess of free dye by a P10 desalting column with a Sephadex G25 matrix (GE Healthcare, Chicago, IL, USA) and concentrated using Amicon Ultra Centricons (Merck, Darmstadt, Germany). Fluorescent OB* and SF were generated by mixing 90 and 10% of unlabeled and labeled αS, respectively. The fluorophore at a position near to 122 was reported to be efficient in following amyloid aggregation and oligomer and fibril stability without affecting much these properties[6]. But in any case, the low ratio of labeled to unlabeled monomers and the C-terminal position of Cys122[8], ensured the absence of significant modifications to the properties of OB* and SF.

**ssNMR**. PRE data were acquired by performing DARR experiments in ssNMR to probe the membrane interactions of the rigid region of the oligomers and fibrils[8].

All lipids were acquired from Avanti Polar Lipids Inc., Alabaster, AL, USA. SUVs of the relevant lipid compositions were prepared by drying chloroform solutions of the lipids under a stream of nitrogen gas followed by desiccation under vacuum. The resulting dried lipid film was then rehydrated in buffer, subjected to 3–5 freeze-thaw cycles, and sonicated until the solution becomes clear. SUVs composed of 1,2-dioleoyl-*sn*-glycero-3-phosphoethanolamine (DOPE), 1,2-dioleoyl-*sn*-glycero-3-phospho-L-serine (DOPS), and 1,2-dioleoyl-*sn*-glycero-3-phosphocholine (DOPC) at a ratio 5:3:2 w/w, respectively, were prepared (this lipid composition has been previously proposed to be a good model that mimics the physiological lipid membrane composition[53]). Two percent of paramagnetic lipid with an unpaired electron located at the hydrophilic head group (1,2-dimyristoyl-sn-glycero-3-phosphoethanolamine-N-DTPA gadolinium salt), or at carbon 16 of the lipid tail (1-palmitoyl-2-stearoyl-[16-doxyl]-sn-glycero-3-phosphocholine) were added to the lipid mixture, to probe the interaction of the protein with the surface of the membrane or the interior of the membrane, respectively. The protein sample was incubated with SUVs for 1 h before pelleting down using a TLA-120.2 Beckman rotor (Beckman Coulter UK Ltd., High Wycombe, UK, 30 min, 300,000 × g). The resulting pellet was transferred into 3.2 mm Zirconia XC thin-walled MAS rotors for ssNMR experiments.

**Interaction of αS species with SUVs using far-UV CD**. To investigate the interaction of αS in the different aggregated states with lipid membranes, we monitored the change in the mean residue ellipticity of αS at 222 nm at increasing lipid:αS molar ratios using SUVS of 5:3:2 of DOPE:DOPS:DOPC. The different αS species (10 µM mass concentration) were incubated with SUVs at varying concentrations for at least 1 h at room temperature prior to measurement, and membrane binding affinity was estimated according to the binding equation derived by assuming a fixed but unknown number of lipid molecules interacting with one molecule of protein with identical affinity[54].

**Calcein release in SUVs**. The kinetics of lipid membrane permeabilisation induced by the different αS species was obtained by monitoring the increase in fluorescence due to the leakage of entrapped calcein within SUVs over time[4]. The different protein species (0.05 µM) were added to SUVs (5 µM) and the fluorescence measurements were obtained using a Varian Cary Eclipse fluorimeter (Palo Alto) at 20 °C in a 2 mm × 10 mm path length cuvette using excitation and emission wavelengths of 485 and 520 nm, respectively. The percentage of calcein release was defined as:

$$\%_{\text{release}} = \left[ \frac{(F - F_{\text{control}})}{(F_{\text{tot}} - F_{\text{control}})} \right] \times 100$$

where $F$ is the fluorescence measured in the presence of the different αS species, $F_{\text{control}}$ is the fluorescence measured of the SUVs alone, and $F_{\text{tot}}$ is the total fluorescence obtained after complete disruption of SUVs using Triton X-100 (1% v/v).

**AFM**. All images were acquired at room temperature in air typically using intermittent contact mode on a Nanowizard II atomic force microscope (JPK Instruments, Berlin, Germany) except for the analysis of OA* and OB* samples, which were acquired using tapping mode in a Multimode 8 atomic force microscope (Bruker, Massachusetts, USA). The different αS species (0.1–1 µM, 10 µL) were applied onto a layer of freshly cleaved mica and allowed to air-dry. The samples were washed with water to remove any salts and dried again before imaging. Images were processed and analyzed with Gwyddion open source software (version 2.48) (http://www.gwyddion.net).

**Far-UV CD**. Far-UV CD spectra of the different αS species were acquired in PBS at 20 °C between 200 and 250 nm, using a scan speed of 50 nm min⁻¹ and a bandwidth of 1 nm. Ten accumulations were recorded for each sample, using a 1-mm path length cuvette and a J-810 Jasco spectropolarimeter (Tokyo, Japan), equipped with a thermostated cell holder.

**FT-IR**. FT-IR spectra of the different αS species (100–400 μM) were acquired in PBS and analyzed in a Bruker BioATRCell II using a Bruker Equinox 55 FT-IR spectrophotometer (Bruker Optics Limited, UK) equipped with a liquid-nitrogen-cooled mercury cadmium telluride (MCT) detector and a silicon internal reflection element (IRE). For each spectrum, 256 interferograms were recorded at 2 cm$^{-1}$ resolution. Data processing of the amide I region (1720–1580 cm$^{-1}$) was performed with the Opus software package (Bruker Optics Limited, UK) and consisted of a background subtraction of the buffer spectra, atmospheric compensation, and baseline subtraction. All absorbance spectra were normalized for comparison.

**X-ray diffraction**. Protein stalks were prepared by air drying 10 μL of ~800 μM (mass concentration) of the different αS species between two wax-filled capillary ends mounted in a Petri dish[55]. X-rays were generated using MICROSTAR microfocus rotating anode X-ray generator and the diffraction data were collected on a X8 Proteum system (Bruker AXS). The data were subsequently analyzed on PROTEUM 2 software suite.

**ThT and ANS fluorescence**. Fluorescence measurements were performed in a 2 mm × 10 mm path length cuvette, using a Varian Cary Eclipse fluorimeter (Palo Alto, CA, USA) in a temperature-controlled cell holder. ThT fluorescence was monitored by exciting the sample at 446 nm and recording the emission fluorescence spectrum between 460 and 600 nm (5-nm slitwidths). Each protein species (10 μM) was incubated with ThT (50 μM, $\varepsilon_{416nm} = 26,620$ M$^{-1}$ cm$^{-1}$) in PBS for 30 min before performing the measurement. ANS binding was monitored by exciting the sample at 350 nm and recording the emission spectrum between 400 and 650 nm (5-nm slitwidths). Each protein species (5 μM) was incubated with ANS (250 μM, $\varepsilon_{350nm} = 5000$ M$^{-1}$ cm$^{-1}$) in PBS for 30 min before recording the spectra.

**Cell cultures**. Authenticated human neuroblastoma SH-SY5Y cells were purchased from A.T.C.C. (Manassas, VA, USA). SH-SY5Y cells were tested negative for mycoplasma contaminations, and were maintained in a 5% CO$_2$ humidified atmosphere at 37 °C and grown until 80% confluence for a maximum of 20 passages[12,37].

Primary cortical neurons were obtained from embryonic day (ED)-17 Sprague-Dawley rats (Harlan)[56], maintained in neuronal basal medium (NBM) at 37 °C in a 5.0% CO$_2$-humidified atmosphere and analyzed 14 days after plating[8,12]. Experiments and animal use procedures were in accordance with the National Institutes of Health Guide for the Care and Use of Laboratory Animals (NIH Publications No. 80-23, revised 1996). The experimental protocols were approved by the "Commissione per l'Etica della ricerca" of the University of Florence, in compliance with the European Convention for the Protection of Vertebrate Animals used for Experimental and Other Scientific Purposes (ETS no. 123) and the European Communities Council Directive of 24 November 1986 (86/609/EEC). The authors further attest that all efforts were made to minimize the number of animals used and their suffering.

Human iPSC-derived dopaminergic neurons were purchased from Axol Bioscience (Cambridge, UK), which obtains human cell resources from cell repositories who guarantee all human cell collections are performed at certified facilities under the highest ethical standards. Discrete legal consent form was obtained and the donors' or clinics' rights to hold research uses, for any purpose, or further commercialization use were waved. All human cells were collected under protocols that are in compliance with the Health Insurance Portability and Accountability Act of 1996 (HIPAA). Human iPSC-derived dopaminergic neuron progenitors were plated and led to maturation according to manufacturer's instructions. In detail, they were plated on 12-well plates containing glass coverslips coated with poly-D-lysine plus surebond-XF solution. Then the maturation to dopaminergic neurons was achieved starting from day 1, by growing the cells in differentiation medium at 37 °C with 5% CO$_2$. On day 5 post-plating, the differentiation medium was replaced with the maintenance medium, which was changed every 2 days. The resulting iPSC-derived dopaminergic neurons were characterized and analyzed between day 14 and day 18 of culturing. The cells were fixed in 4% (v/v) paraformaldehyde at room temperature and blocked with fetal bovine serum in 0.1% Triton X-100 for 30 min. Coverslips were incubated with 1:300 diluted rabbit anti-MAP-2 antibodies (ab32454, Abcam) and 1:200 diluted mouse anti-TH antibodies (sc-25269, Santa Cruz Biotechnology) in blocking solution overnight at 4 °C. After washing with PBS, the cells were incubated with 1:500 diluted Alexa-Fluor-568-conjugated anti-rabbit secondary antibodies (Thermo Fisher Scientific) and 1:500 diluted Alexa-Fluor-514-conjugated anti-mouse secondary antibodies (Thermo Fisher Scientific) for 90 min at room temperature. Finally, coverslips were incubated in PBS containing DAPI for 15 min at room temperature. Fluorescence emission was detected after double excitation at 568 and 514 nm by a TCS SP8 scanning confocal microscopy system (Leica Microsystems, Mannheim, Germany).

**Confocal microscopy analysis for the penetration of αS species**. αS species were added to the cell CM of SH-SY5Y cells seeded on glass coverslips typically for 1 h at 0.3 μM. In a set of experiments, OB*, SF, or LF were added to the cell CM of SH-SY5Y cells seeded on glass coverslips for different lengths of time (0, 1, 3, 6, and

24 h). In another set of experiments, SH-SY5Y cells were treated for 6 h at 37 °C and then with or without 0.05% trypsin for 15 min at 4 °C. In an additional experiment, SH-SY5Y cells were treated for 6 h at 37 °C with or without 5 μM dynamin inhibitor I, dynasore (Sigma-Aldrich, St. Louis, MO, USA) or for 6 h at 4 °C. Following membrane permeabilization, αS was detected with 1:250 diluted rabbit polyclonal anti-αS antibodies (ab52168 Abcam, Cambridge, UK) and with 1:1000 diluted Alexa-Fluor-488-conjugated anti-rabbit secondary antibodies (Thermo Fisher Scientific). In another set of experiments, OB* and SF were added to the CM of SH-SY5Y cells seeded on glass coverslips for 1 h at increasing concentrations (0.03, 0.1, 0.3, 1.0, and 3.0 μM) and the immunostaining analysis was performed as described below.

In a set of experiments, OB*, SF, and LF at 0.3 μM were added to the CM of iPSC-derived dopaminergic neurons for 24 h. The cells were fixed in 4% (v/v) paraformaldehyde at room temperature and blocked with fetal bovine serum in 0.1% Triton X-100 for 30 min Coverslips were incubated with 1:400 diluted mouse anti-MAP-2 antibodies (ab11267, Abcam) and 1:250 diluted rabbit anti-oligomer A11 polyclonal antibodies (AHB0052 Thermo Fisher Scientific) in blocking solution overnight at 4 °C. After washing with PBS, the cells were incubated with 1:500 diluted Alexa-Fluor-568-conjugated anti-rabbit secondary antibodies (Thermo Fisher Scientific) and 1:500 diluted Alexa-Fluor-514-conjugated anti-mouse secondary antibodies (Thermo Fisher Scientific) for 90 min at room temperature.

In another set of experiments, OB*, SF, and LF at 0.3 μM were added to the CM of SH-SY5Y cells for different lengths of time (0, 3, 6, 14, and 24 h). Following membrane permeabilization, αS was detected with 1:250 diluted rabbit anti-oligomer A11 polyclonal antibodies (AHB0052 Thermo Fisher Scientific), and with 1:1000 diluted Alexa-Fluor-488-conjugated anti-rabbit secondary antibodies (A-11034 Thermo Fisher Scientific). In another set of experiments, OB* and SF were added to the CM of SH-SY5Y cells at 0.3 μM for different lengths of time (0, 3, 6, 14, and 24 h). Following membrane permeabilization, αS was detected with 1:300 diluted rabbit oligomer-specific Syn33 polyclonal antibodies (ABN2265M, Sigma-Aldrich) and then with 1:1000 diluted Alexa-Fluor-488-conjugated anti-rabbit secondary antibodies (A-11034 Thermo Fisher Scientific).

In another set of experiments, OB* were added to the CM of SH-SY5Y cells at different concentrations (0.03, 0.1, 0.3, 1, and 3 μM) for 1 h. αS was detected with 1:250 diluted rabbit anti-oligomer A11 polyclonal antibodies (AHB0052 Thermo Fisher Scientific), and with 1:1000 diluted Alexa-Fluor-488-conjugated anti-rabbit secondary antibodies (A-11034 Thermo Fisher Scientific). In another set of experiments, OB*, SF, and LF at 0.3 μM were added to the CM for 30 min and then polyclonal A11 (AHB0052 Thermo Fisher Scientific) or OC (AB2286 Sigma-Aldrich) conformation-sensitive antibodies were added for 24 h to the extracellular medium (in a molar ratio of 1:2.5). The cells were then counterstained with 5.0 μg/ml Alexa-Fluor 633-conjugated WGA (W21404 Thermo Fisher Scientific)[57] fixed with 2% (v/v) paraformaldehyde, permeabilized and αS was detected with 1:250 diluted rabbit polyclonal anti-αS antibodies (ab52168 Abcam) and with 1:1000 diluted Alexa-Fluor-488-conjugated anti-rabbit secondary antibodies (A-11034 Thermo Fisher Scientific). Fluorescence emission was detected after double excitation at 633 and 488 nm by the TCS SP8 scanning confocal microscopy system (Leica Microsystems, Mannheim, Germany) equipped with an argon laser source. A series of 1.0-μm-thick optical sections (1024 × 1024 pixels) was taken through the cell depth for each sample using a Leica Plan Apo ×63 oil immersion objective and projected as a single composite image by superimposition. In some experiments, four optical sections near the coverslip were merged to a single composite image, which we referred to as basal planes, four optical sections more distant from the coverslip were merged and referred to as median planes and four higher top optical sections were merged and referred to as apical planes. The confocal microscope was set at optimal acquisition conditions, e.g., pinhole diameters, detector gain, and laser powers. Settings were maintained constant for each analysis.

**Measurement of intracellular ROS and mitochondrial superoxide ions**. αS species were added to the CM of SH-SY5Y cells seeded on glass coverslips for 15 min at various concentrations (0.03, 0.1, 0.3, 1.0, and 3.0 μM). In a set of experiments, 0.3 μM OB* were added to the CM for different lengths of time (0, 5, 10, 15, 30, and 60 min). In a set of experiments, LPS-free αS species (M, OB*, and LF) were compared to normal samples, by adding them to the CM of SH-SY5Y cells seeded on glass coverslips for 15 min at 0.3 μM. We used this time the CM-H$_2$DCFDA probe to detect and quantify intracellular levels of hydrogen peroxide as a ROS marker, although we reported similar results for the OB* when the intracellular accumulation of ROS was followed by Hyper-3 (probe for hydrogen peroxide) and dihydroethidium (probe for superoxide radical) in primary neuronal cells[39]. In order to quantify hydrogen peroxide, then, cells were loaded with 5 μM 2′,7′-dichlorodihydrofluorescein diacetate for 10 min (CM-H$_2$DCFDA, Thermo Fisher Scientific)[8,45]. The emitted fluorescence was detected at 488 nm excitation line by the confocal scanning system described above.

Mitochondrial superoxide ion production was detected with a static time-point measurement in living SH-SY5Y cells with MitoSOX probe (Thermo Fisher Scientific). Fluorescently labeled AF488-OB* and AF488-SF were added to the CM of SH-SY5Y cells for 1 h at 0.03, 0.1, and 0.3 μM, and the red and green emitted fluorescences were detected at 550 and 488 nm excitation lines, respectively, by the

confocal scanning system described above. As a positive control, cells were also treated with 250 μM $H_2O_2$ and the analysis was performed as described above.

**Alteration of membrane permeability**. The membrane integrity disruption was assessed in SH-SY5Y cells and primary rat cortical neurons seeded on glass coverslips[57]. Briefly, cells were loaded with 1.0 μM calcein-AM (Thermo Fisher Scientific) for 10 min at 37 °C and then treated for 1 h with αS species at 0.3 μM. In another set of experiments, 0.3 μM OB* were added to the CM of SH-SY5Y cells for different lengths of time (0, 5, 10, 15, 30, and 60 min). The membrane integrity disruption was also assessed in SH-SY5Y cells treated for 1 h with αS species (M, OA*, and OB*) derived from N-acetylated αS protein at 0.3 μM. The emitted fluorescence was detected at 488 nm excitation line by the confocal scanning system described above.

**Measurement of intracellular $Ca^{2+}$**. αS species were added to the CM of SH-SY5Y cells and primary rat cortical neurons seeded on glass coverslips for 15 min at various concentrations (0.03, 0.1, 0.3, 1.0, and 3.0 μM). In a set of experiments, 0.3 μM OB* were added to the CM of SH-SY5Y cells for 0, 5, 10, 15, 30, 60, and 180 min and to primary rat cortical neurons for 0, 5, 15, 60, and 180 min. Cells were also treated with 0.3 μM OB* in CM without $Ca^{2+}$. In a set of experiments, cells were treated for 15 or 180 min with 0.3 μM M in the absence or presence of 0.03 μM SF (monomers equivalents, corresponding to 10% of monomers). Then the cells were loaded with 10 μM Fluo-4 AM (Thermo Fisher Scientific) and the analysis was performed by confocal microscopy (excitation at 488 nm)[37,57].

In a set of experiments, the $Ca^{2+}$ influx was analyzed in real-time in living SH-SY5Y cells loaded with the Fluo-4 AM probe for 10 min. The intracellular $Ca^{2+}$ basal level in living cells was measured for 10 min and then $Ca^{2+}$ currents were analyzed following the addition of OB* or SF up to 30 min. The emitted fluorescence was detected at 488 nm excitation line over time by the confocal scanning system described above.

**Measurement of caspase-3 activity**. OB*/SF/LF were added to the CM of iPSC-derived dopaminergic neurons seeded on glass coverslips for 24 h at 0.3 μM. In another set of experiments, SH-SY5Y cells seeded on glass coverslips were treated with M/OA/OB*/SF/LF for 24 h at 0.3 μM or for different lengths of time (0, 1, 3, 5, and 24 h). After incubation, the CM was removed and replaced for 1 h with the FAM-FLICA Caspase-3/7 solution (Immunochemistry Technologies, LLC, Bloomington, MN). The FLICA reagent FAM-DEVD-FMK enters each cell and binds irreversibly and covalently to active caspase-3. The reagent bound covalently to the active enzymes is retained in the cell, while the unbound reagent diffuses outside and is washed away. The emitted fluorescence was then detected at 488-nm excitation by the confocal scanning system described above.

**MTT reduction assay**. The cytotoxicity of αS species was assessed on SH-SY5Y cells and primary rat cortical neurons seeded in 96-well plates, 24 h after their addition to the CM at various concentrations (0.03, 0.3, and 3.0 μM), by the MTT assay[37,58]. SH-SY5Y cells were treated for 24 h with αS species (M, OA*, and OB*) derived from N-acetylated αS protein at 0.3 μM. In a set of experiments, LPS-free αS species (M, OB*, and LF) were compared to normal samples, by adding them to the CM of SH-SY5Y cells seeded on glass coverslips for 24 h at 0.3 μM. In a set of experiments, αS species were added to the CM of SH-SY5Y cells at 0.3 μM for different lengths of time (0, 1, 3, 5, and 24 h) and the MTT assay was assessed. In another set of experiments, OB*, SF, and LF at 0.3 μM were added to the CM of SH-SY5Y cells and, following 30 min of treatment, A11 or OC antibodies (in a molar ratio of 1:2.5) (AHB0052 Thermo Fisher Scientific and AB2286 Sigma-Aldrich, respectively), were added to the extracellular medium for 24 h and the MTT assay was assessed. A11 and OC antibodies alone were also analyzed as control. After treatment, the CM was removed, cells were washed with PBS and the MTT solution was added to the cells for 4 h. The formazan product was solubilized with cell lysis buffer (20% sodium dodecyl sulfate (SDS), 50% N, N-dimethylformamide, pH 4.7) for 1 h. The absorbance values of blue formazan were determined at 590 nm. MTT tests were achieved using Microplate Manager® Software (Biorad, CA, USA). Cell viability was expressed as the percentage of MTT reduction in treated cells as compared to those untreated.

**Dot-blot analysis**. Dot-blot analyses of αS species (OA, OB*, SF, and LF) were performed by spotting 2.0 μl (0.36 mg/ml) of each conformer onto a 0.2 μm nitrocellulose membrane. After blocking (1.0% bovine serum albumin in TBS/TWEEN 0.1%) the blots were probed with 1:1200 diluted rabbit anti-oligomer A11 polyclonal antibodies (AHB0052, Thermo Fisher Scientific), or with 1:1000 rabbit anti-amyloid fibrils OC (AB2286, Sigma-Aldrich) or with 1:1250 diluted conformation-insensitive rabbit polyclonal anti-αS antibodies (ab52168 Abcam) or with 1:250 diluted conformation-insensitive mouse monoclonal 211 anti-αS antibodies (sc12767, Santa Cruz Biotechnology). In a set of experiments, αS species (OB*, SF, and LF) spotted onto the nitrocellulose membrane were detected with 1:3000 diluted rabbit oligomer-specific Syn33 antibody (ABN2265M, Sigma-Aldrich). Then, the blots were incubated with 1:3000 diluted HRP-conjugated anti-rabbit or anti-mouse secondary antibodies (AB6721 and AB6728, Abcam). The immunolabeled bands were detected using a

SuperSignalWest Dura (Pierce, Rockford, IL, USA) and ImageQuant™ TL software (GE Healthcare UK Limited version 8.2).

**Confocal microscopy analysis of αS species in the absence of cells in vitro**. SF and LF at 0.3 μM were incubated in CM without cells in wells containing a glass coverslip for 0, 1, 3, and 24 h at 37 °C. Following the incubation, the coverslips were fixed with 2% (v/v) paraformaldehyde, incubated with BSA 0.5% to avoid unspecific adherence of antibodies to the glass coverslip, and finally incubated for 30 min at 37 °C with 1:2000 diluted mouse monoclonal 211 anti-αS IgG1 antibodies (sc12767, Santa Cruz Biotechnology) or with 1:2500 diluted rabbit anti-oligomer A11 polyclonal antibodies (Thermo Fisher Scientific), and then for 30 min with 1:2000 Alexa-Fluor 514-conjugated anti-mouse or anti-rabbit secondary antibodies (Thermo Fisher Scientific). The emitted fluorescence was detected at 514-nm excitation line by the confocal scanning system described above. The analysis was also performed with OB* at 0 and 24 h as positive control, and with primary and secondary antibodies without αS, to exclude any cross-reaction of the antibodies.

**DLS**. Size distribution analysis was performed with a Malvern ZetasizerNano S DLS device (Malvern Panalytical, Malvern, United Kingdom) using SF at 1 μM in PBS following incubation at 37 °C for different lengths of time (0, 3, 14, and 24 h). OB* were also analyzed at 1 μM in PBS following incubation at 37 °C for 0 and 24 h. Each sample was analyzed considering the refraction index and viscosity of its dispersant. A 45-mm reduced volume plastic cell was used.

**Analysis of the kinetics of toxicity using the different cellular readouts**. The different intracellular fluorescence intensities associated with $Ca^{2+}$ influx and MTT reduction were plotted versus the time elapsed after αS addition to the CM. A variable number of cells were analyzed for every probe and time point and the resulting kinetic plots were analyzed with a procedure of best-fitting using a single exponential function of the form:

$$F(t) = F(\text{eq}) + A \exp(-kt) \qquad (1)$$

where $F(t)$ is the intracellular fluorescence at time $t$ as a percentage of that observed in untreated cells, $F(\text{eq})$ is the same fluorescence at the apparent equilibrium (time ∞), $A$ is the amplitude of the exponential fluorescence change as a percentage of that observed in untreated cells, and $k$ is the apparent rate constant in $s^{-1}$.

The intracellular fluorescence intensities associated with the intracellular αS species recognized by A11 antibody were plotted versus the time elapsed after αS addition to the CM and the resulting kinetic plots were analyzed with a procedure of best-fitting using a sigmoidal function of the form

$$F(t) = F(\text{eq}) + \frac{F(0) - F(\text{eq})}{1 + \left(\frac{kt}{A}\right)^B}$$

where $F(t)$ is the intracellular fluorescence at time $t$ as a percentage of that observed in untreated cells, $F(0)$ is the same fluorescence at time zero, $F(\text{eq})$ is the same fluorescence at the apparent equilibrium (time ∞), $A$ is the amplitude of the fluorescence change as a percentage of that observed in untreated cells, $k$ is the apparent rate constant in $s^{-1}$, and $B$ is the slope of the sigmoidal function at time $t$.

**STED microscopy**. STED xyz images (i.e., z-stacks acquired along three directions: x, y, and z axes) of rat cortical neurons treated with OB* and SF at 0.3 μM for 14 and 24 h were acquired by using an SP8 STED 3X confocal microscope (Leica Microsystems, Mannheim, Germany)[37]. Primary rat cortical neurons were counterstained with 0.01 mg/ml WGA, Tetramethylrhodamine Conjugate (W849, Thermo Fisher Scientific). αS was detected with 1:125 diluted conformation-insensitive mouse monoclonal 211 anti-αS IgG1 antibodies (sc12767, Santa Cruz Biotechnology) that recognize only the human protein, and 1:500 Alexa-Fluor 514-goat anti-mouse IgG1 secondary antibody (A-31555, Thermo Fisher Scientific). Fluoromount-G™ (00-4958-02, Fisher Scientific) was used as mounting medium. Fluorescence emission was detected after double excitation at 550 and 514 nm. STED xyz images were acquired in bidirectional mode with the Leica SP8 STED 3X confocal microscope. Tetramethylrhodamine fluorophore was excited with a 550-nm-tuned white light laser (WLL) and emission collected from 564 to 599 nm, Alexa-Fluor 514 was excited with a 510-nm-tuned WLL and emission collected from 532 to 551 nm. Frame sequential acquisition was applied to avoid fluorescence overlap. It was applied a gating between 0.3 and 6 ns to avoid collection of reflection and autofluorescence. Six-hundred-and-fifty-nanometer pulsed-depletion laser was used for Alexa-Fluor 514 excitation. Images were acquired with Leica HC PL APO CS2 100x/1.40 oil STED White objective. Gated pulsed-STED was applied to Alexa-Fluor 514 fluorophore. Collected images were de-convolved with Huygens Professional software (Scientific Volume Imaging B.V., Hilversum, The Netherlands; version 18.04) and analyzed with Leica Application Suite X (LAS X) software (Leica) to generate 3D reconstructions. Z-series stacks were obtained from 5 μm neuron slices. Images were collected at 0.1-μm intervals.

In another experiment, SH-SY5Y cells were treated with fluorescently labeled AF488-OB* and AF488-SF for 24 h and STED xyz images were acquired as described above after double excitation at 550 and 488 nm.

**Measurements of the fluorescence intensities**. To quantify the intensity of the fluorescent signal arising from each probe, cells were analyzed by using the ImageJ software (NIH, Bethesda, MD; version 1.52t). The fluorescence intensities were measured in regions of interest, centered on single cells, whose dimensions remained constant for all the analyzed images, and were then expressed as the ratio ($\Delta F/F$) of the change in fluorescence ($\Delta F$) with respect to the resting baseline fluorescence in untreated cells ($F$)[57,59].

In immunofluorescence experiments, intracellular αS fluorescence intensity was assessed in median planes by selecting multiple ROIs per cell in the cytoplasmic compartment, excluding the cellular membrane, after background subtraction. Similarly, the extracellular fluorescence was measured in basal, median, and apical planes by selecting multiple ROIs per cell in the membrane compartment excluding the cytoplasmic one, in order to include only the signals arising from the species interacting with the outer leaflet of cell membranes. The fluorescence intensities were then expressed as fractional changes above the resting baseline for untreated cells ($\Delta F/F$). In a set of experiments, the intracellular fluorescence was analyzed by counting the number of puncta inside the cells, by using the ImageJ software. In STED microscopy analyses, the fluorescence intensity arising from intracellular and extracellular αS was expressed in arbitrary units, after background subtraction.

**Comparison of the concentration of the fibrils**. We estimated the same fibril number concentrations for SF and LF (corresponding to a higher mass concentration for LF). For OB*, we took the average number of monomers per oligomer from AUC analysis[4]. This was ca. 30 monomers. For the fibrils we can estimate the number of monomers per fibril using the following equation:

$$N = \frac{\rho V N_A}{MW} = \frac{\rho \pi \left(\frac{h}{2}\right)^2 l \, N_A}{MW}$$

where $N$ is the number of monomers per aggregate, $N_A$ is the Avogadro's constant, MW is the molecular weight of the monomeric components, $p$ is the density of the aggregate (1.35 g/cm$^3$), $V$ is the volume of the fibril, and $h$ and $l$ are the height and length of the fibril, respectively (derived from AFM). The numbers obtained are ca. 60 monomers for SFs and ca. 500 monomers for LFs.

Thus, 0.3 μM concentration (monomer equivalents) of SF and 3 μM concentration (monomer equivalents) of LF give rise to ca. 0.005 μM particle concentration in both samples.

**Statistical analysis**. All data were expressed as means ± standard error of mean (S.E.M.). Comparisons between the different groups were performed by ANOVA followed by Bonferroni's post comparison test, by using GraphPad Prism 7.0 software.

**Reporting summary**. Further information on research design is available in the Nature Research Reporting Summary linked to this article.

## Data availability

All data supporting the findings of this study are provided within the paper and its supplementary information. A source data file is provided with this paper. All additional information will be made available upon reasonable request to the authors. The data reported in Fig. 1a are deposited at the placeholder DOI link: https://doi.org/10.17863/CAM.59871. Source data are provided with this paper.

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

## Acknowledgements

We thank Maria Lucia Angelotti, Tommaso Staderini, Emilio Ermini, Janet Kumita, and Alfonso De Simone for their technical support and scientific insight. This research was supported by the University of Florence (Fondi Ateneo to F.C. and C.C.), the Ministry of Education, Universities and Research of Italy (Progetto Dipartimento di Eccellenza "Gender Medicine" to C.C.), Parkinson's UK (G-1508 to S.W.C. and C.M.D.), the Center for Misfolding Diseases of the University of Cambridge (S.W.C. and C.M.D.), the UK Medical Research Council (MR/N000676/1 to C.M.D.), the Agency of Science, Technology and Research of Singapore (to S.W.C.), and the Ministry of Economy and Competitiveness of Spain (MINECO RYC-2012-12068 and MINECO/FEDER EU BFU2015-64119-P to N.C.).

## Author contributions

R.C. and A.B. performed the cellular experiments and R.C. carried out the associated data analysis. S.W.C. and J.D.C. performed the AFM measurements and data analysis. S.W.C. and C.X. performed the remaining in vitro experiments and the associated data analysis. R.C., F.C., N.C., and C.C. were principally involved in the design of the study. R.C., N.C., C.M.D., F.C., and C.C. wrote the paper, all authors were involved in the analysis of the data and editing of the manuscript.

## Competing interests

The authors declare no competing interests.
