## [Peer Review File · Nature Communications]

Reviewers' Comments:

Reviewer #1:

Remarks to the Author:

This is an interesting study detailing a mechanism of cell toxicity, i.e. the release of toxic oligomers from alpha-synuclein (a-syn) fibrils bound to the cell membrane, of likely relevance to disease pathogenetic processes. Important issues should be addressed, however, to strengthen interpretation of the results and legibility of the paper. Quite importantly, some of the conclusions of the manuscript do not seem to be in line with the results presented and should therefore be reformulated or better justified.

(1) The last sentence of the abstract needs to be clarified. The sentence indicates that a-syn fibrils are responsible for neuron-to-neuron spreading of a-syn pathology. However, cell-to-cell passage of a-syn is not investigated as part of this current study and in fact, as further indicated below, the present findings do not really support the likelihood that fibrils, if released in the extracellular space, may be easily taken up by other cells.

(2) The initial sets of experiments were performed using synthetic membranes. From these experiments, it is concluded that (i) the beta-sheet cores of short fibrils (SF) and long fibrils (LF) do not interact significantly with lipid bilayer, (ii) toxic oligomeric species (OB*), SF and LF are all able to interact with synthetic vesicles, and (iii) OB* is particularly effective in disrupting membrane integrity. Taken together, these findings probably indicate that fibrils may interact with the membrane surface but do not cause significant disruption of the lipid bilayer. This conclusion, if correct, should be clearly stated.

(3) In general, the description of the results should be made clearer. While reading the paper, data description in the text cannot always be easily correlated to the corresponding figures. It also appears that reference to specific figures may be erroneous. For example, reference is made to Figure 2C (instead of Figure 2B) after the statement "70%, 12% and 6% of the OB*, SF and LF samples were internalized...." (page 6).

(4) Data in Figures 2B, C and D are reported as percent of endogenous a-syn. Did endogenous a-syn remain stable or did it change after treatment with OB*, SF or LF?

(5) Data on the internalization of OB*, SF and LF are reported after incubations for 1h (page 6). Do features of internalization of OB*, SF or LF change after longer incubations?

(6) The use of N-acetylated OB* should be explained/justified.

(7) "Taken together our results indicate that a-syn short fibrils..." (page 8). This is an important summary of the results on internalization and toxicity of a-syn fibrils. Why no reference is made in this summary to the results obtained with long fibrils? Were the results obtained with SF and LF overall consistent?

(8) A critical set of experiments in support of the main conclusion of this paper (i.e. that a-syn fibrils release oligomers that are ultimately responsible for their toxicity) is carried out with the conformation-sensitive antibody A11. Other conformation-sensitive antibodies are available and, given the importance of these experiments, validation of some of the results obtained with A11 using at least one more conformation-sensitive antibody is warranted.

(9) The time points at which analyses were made should be justified and, if needed, made more consistent. For example, visualization of fibrils and oligomers using STED microscopy was carried out after incubations for 14 h. Why was this time point chosen whereas, for other experiments, analyses were made after 24h?

(10) An important control experiment involved the addition of monomers to SF or LF in order to evaluate if formation of oligomers in the extra-cellular space would result in their internalization and ultimately their toxicity (in the form of calcium influx). Cells were incubated with SF or LF in the presence or absence of monomers for 15 min or 180 min. These incubation times may not be sufficient for the generation of oligomers, their internalization and the production of toxic effects. Please explain/justify.

(11) Images shown in Figure 6A are not very clear in representing the effect of A11 or OC on the internalization of a-syn after incubations with SF and, particularly, LF.

(12) The ability of A11 to block a-syn internalization after exposure to SF and LF is an important finding. Is this finding due to the fact that this antibody binds oligomeric species that are first released in the extracellular space and then taken up by cells? Please explain/interpret.

The conclusion that fibrillar a-syn is responsible for neuron-to-neuron spreading of a-syn (page 14) is not supported by any of the data shown in this paper. In fact, findings of this study obtained with SH-SY5Y cells indicate that only 10% or less of fibrils added to the culture medium were actually internalized; most of these fibrils remained bound to the membrane but were not taken up (page 6). Data in Figure 4C also suggest that, with SF, very little or no significant amounts of fibrils gained access into the cytosolic space. Finally, data in Figure 5 clearly show lack of internalization of a-syn fibrils. How would these data be compatible with neuron-to-neuron transfer of a-syn fibrils? These data seem inconsistent with previously published findings showing uptake of a-syn fibrils in vitro (please explain). Overall, the present results strongly support a key role of toxic oligomers in pathogenetic processes (please comment). Is it possible that these oligomers, once generated and internalized from fibrils bound to the cell membrane, could also play an important role in neuron-to-neuron transfer of a-syn pathology?

Reviewer #2:

Remarks to the Author:

General Comments:

In this manuscript, the authors define the nature of α -synuclein species that induce neuronal damage, and in particular describe a mechanism by which short fibrils may release oligomeric species that induce neuronal dysfunction. Overall this is interesting to read, and demonstrates a new potential phenomenon that would be important to the field. The questions raised below are both conceptual for the selection of model and approach to characterise the species inside cells, and technical with regards to the experimental paradigms.

Many of the results shown here have been already demonstrated by this group, and others, in a range of cell systems and in vivo models: namely that the oligomeric species either recombinant OB, or endogenously generated, can exert toxicity, through induction of ROS and oxidative stress, membrane perturbation, and cell death. The demonstration of some of these well-established features of synucleinopathy mostly in a neuroblastoma cell line using recombinant species, and the significant new mechanistic advances, over the published phenotypes, was not clearly described. The relevance of this model to the aggregation of alpha-synuclein, and the choice for this approach over existing transgenic animal or human models is not clearly articulated. The kinetics of the internalisation, the kinetics of aggregation and disaggregation and oligomer release, and the mechanisms of induction of ROS and membrane perturbation, and even toxicity in a neuroblastoma cell line are likely to differ significantly from human neurons, and therefore will the results here be relevant for human brain and disease: it would be improved significantly if these effects were present in other in vitro or in vivo models of PD.

The finding of the presence of the toxic oligomeric species on addition of SFs is the most novel

finding of this study, and very interesting to the field. The data for this rests on observations made by quantifying fluorescence from immunocytochemistry utilising antibody based tools that are difficult to be certain of their specificity for structural species within cells. Orthogonal biophysical approaches in cells that can prove this hypothesis would greatly add to this work.

The authors provide a semi-quantitative assessment and correlation of different structural forms and their effects in cells. Their comparison of one species vs another is based on the number of monomer equivalents in any aggregate. Yet in the discussion they speculate that the toxic aspect is due to the ends of any fibril, suggesting that the number of ends is the relevant parameter for toxicity. In that case, they would need to make equivalent the concentration of fibrils (numbers/volume) rather than the monomer equivalents, otherwise it is not possible to infer results between SF and LF (in which the LFs would be fewer in number as more monomer equivalents, and therefore fewer open ends).

There is a lack of description of the methods used to characterise the samples, and key details such as the purity of the samples for the species of interest. Whilst this is referenced, the nature of the species and their heterogeneity and variability is key here where biological effects are measured, and should be included. There is also a lack of description of the methods for the image analysis that derives % values – for example the image analysis pipeline that generates the values for intracellular, extracellular and total α -synuclein; and whether these are intensity or puncta values, and how the normalisation of this data occurred.

Specific comments:

Characterisation of species

Why and how are the kinetics of calcein release accounted for by the structure of the SF vs LF. How many aggregates are at the lipid membrane, and how transient is the disruption? What is mechanism of membrane disruption in OB vs SF vs LF? Is it that more aggregates insert or that for any one aggregate, it induces greater disruption? This would be relevant information for understanding the effect of OB vs SF vs LF in cells.

Interaction and internalised species:

There is a lack of detail of this analysis: what is measured here as the morphology of the signal as well as intensity varies with species? (Intensity of fluorescence on ICC is not normally a reliable quantifiable variable). Why use the ratio of one plane to another in the cell, rather than analysing the fluorescence through the volume of a z stack? The control data for this analysis approach is not shown. This could be not possible in polarised cells such as neurons for example (and there is no clear evidence that α -syn is evenly distributed through the cell on uptake). What is the threshold set for endogenous α -syn? Is it the same for all samples and cells? How is intracellular α -syn measured vs extracellular?

Why is there no monomer or OA uptake? Many papers have added monomers to cells and shown biological effects, and monomeric uptake is well documented.

Fig 2 C,D: 'Bell shaped curve for intracellular and total therefore the amount of oligomers internalised into cells directly proportional to those interacting with membrane' How is this result inferred or measured? How did the authors determine the fluorescence signal arising from the plasma membrane?

In the images there is minimal overlap of membrane and α -synuclein so do the authors presume that the traversing across by endocytosis is the %age that interacted with membrane? How did they decide this assumption is true – is this referenced?

A correlation of ROS using live imaging, and intracellular protein level by ICC is clearly demonstrated. Proof of this would be to measure the amount of internalised α -synuclein in the same experiment as the ROS measurement for confirmation of the correlation.

Calcein leakage from SHSY5Y cells, or calcium influx experiments:

What does a basal calcein reduction mean biologically? Is the membrane disruption constant and aggregates are remaining in the membrane? (This is not apparent on confocal imaging.) or do the aggregates pass through membranes and leave the disruption stably?

Calcium influx should be compared to the same cell at basal level rather than 'untreated cells', as otherwise there is no control for the loading of the dye which is different in different cells

Calcium influx in neurons is transient and spikes related to transient binding of aggregates to membranes, therefore how does a snapshot at a fixed timepoint capture this in the rodent neurons. Calcium imaging in neurons would require dynamic imaging and an analysis of the number of spikes and their amplitude, unless there is clearly a change in basal fluophore signal and a basis for this?

What are the kinetics of calcium influx for SHSY5Y cells: is it the same as for calcein reduction? Where is the control in calcium free media?

The MTT assay is a crude measure of viability, based on mitochondrial metabolism but not a specific indicator of mitochondrial dysfunction, which appears overstated. Why isn't the caspase 3 assay performed in neurons? Where is proof of membrane disruption or ROS being actually toxic to neurons where the mechanism is relevant? And how do these fairly different effects in cells interact to cause toxicity?

Results 4:

Characterisation of the A11 antibody with dot blot analysis looks very promising, but does not inform what A11 detects in cell lysates, or in ICC? Does it recognise other oligomers or aggregates in cells, and can that data be generated?

Fig 4B – how is there no background in cells using a total α -synuclein antibody – is this adjusted in the images, or is it true finding?

Did the authors detect the presence of oligomers on adding SF and LF to neurons as well as neuroblastoma cells?

Why not perform this experiment in the absence of cells to demonstrate that oligomer release occurs without cells? In that case the dot blot characterisation of the A11 is relevant and the kinetics of release of oligomers can be determined far more accurately.

If this does not occur in vitro in cell free systems, then what is the cellular mechanism that causes this?

Results 5:

STED imaging – it was not clear why the authors are not using the A11 antibody to measure oligomers in these experiments?

There is only qualitative analysis here, why no quantification of this data, or presentatio of the precision and resolution of the imaging.

The images with scale bar show that the fluorescent spots are roughly 500nm in size, much larger than diffraction limit of light and therefore not at super resolution. In these conditions, how does a spherical aggregate resemble an oligomer or a circular group of fibrils??

I did not follow how this therefore showed that the A11 fluorescent spots in previous experiments, and the 211 AB fluorescent signals both relate to exogenous α -syn (as the expts with A11 happen over 24hrs and so may easily recruit endogenous α -syn)

If release of α -syn monomer by SF and secondary nucleation can occur, why not test this first in

vitro rather than in cells?

Results 6:

Addition of the A11 antibody – if primarily recognises only OB, how does it prevent SF and LF internalisation by 85% (ie even more than OB internalisation). Prevention of internalisation of the species recognised by A11 (OB but also 85% SF and LF) abolished toxicity, in support of the previous experiments that prove that the OB is the toxic form. The OC antibody is then used to show that prevention of internalisation of SF and LF, but not OB, abolishes toxicity. This is indirect evidence to support that SF and LF cause toxicity by oligomeric release, but how is the specificity of the OC antibody determined in cells?

Reviewer #3:

Remarks to the Author:

Summary:

This manuscript reports that a particular type of alpha-synuclein (aS) oligomer, OB*, displays a number of properties potentially related to toxicity and that short fibrils shed these oligomers from their ends. A lot of data is presented characterizing the oligomers and their properties, but the novel observation is the release of toxic oligomers from the ends of fibrils. While this is a novel and very interesting and significant idea of broad interest, the data on this and some of the other issues could be clearer and more convincing.

Major points:

1. Under the heading "aS fibrils gradually release oligomers that are ultimately responsible for their toxicity", the authors claim that short fibrils (SF), prepared by sonication of long fibrils (LF) release toxic oligomers. In Fig. 4B, they show a time dependent increase in A11 immunoreactivity for SF incubated in cell culture medium by confocal microscopy, but it isn't clear that this is a "release" of oligomers from SF or a conformation change in SF exposing the A11 epitope. Clearly, something is going on, but it isn't clear that it is release.

Similarly, the experiment in 4C is to incubate SF with cells and then perform western blotting on the membrane and cytosol fraction with the aS antibody 211. There is a difference for the SF treated samples between membrane and cytosol fraction in terms of intensity of staining being stronger in the membrane fraction and the pattern of bands is different between SF and OB, but I don't see any results that are consistent with the interpretation "release of oligomers". Key controls are missing, like what the preparations look like without incubation with cells and whether oligomers are "released in the absence of cells or SDS.

In order to demonstrate release, you would need to show separation of a smaller oligomer from its parent SF. If it is release from the ends, then the rate should be proportional to the number of ends, so you would expect LF to do it too but at a slower rate. LF were not examined in 4B and C. There should be some means of characterizing the size of the released oligomer, like sedimentation velocity or gel filtration. Additionally, a more detailed kinetic study of the release would be a significant improvement. Right now there is only one timepoint.

2. Another issue is whether the aS oligomers and fibrils have been internalized, where they are localized and how they are internalized. How do you know they are intracellular? How do you know they cross the membrane, as opposed to being endocytosed? Endocytosis can be blocked pharmacologically and by temperature. The methods are not clear on these points, but it looks like the authors are just going by their subjective interpretation of images presented. This would not allow the localization to be determined with the precision needed to make their interpretation. It would be more convincing to show that the fraction that is presumed surface bound can be removed by washing at low pH or protease digestion. Alternatively, the aS could be colocalized with markers of different compartments in the endocytic pathway. TIRF microscopy would be more convincing of membrane penetration and the fact that the amyloid is crossing the membrane

although it is not able to see deep within the cytosol. It might be better to count puncta rather than the total fluorescence signal, since the interesting fraction is aggregated. There is quite a difference in terms of the number of particles that are observed in the nucleus and it would be easier to count this unambiguously (the nucleus is distinct from the "cytosol"). The significance of nuclear synuclein aggregates is not clear. I am not aware of any reports of them in human samples.

Minor points:

1. Introduction. There is considerable discussion about fibrils SF and LF and oligomers, but these terms are ill defined, both here and in the literature in general. What is the difference between an oligomer and fibril? What is the cut off for something to be called oligomeric or short fibril vs long fibril? Is there a structural definition that makes more sense than OA, OB, SF and LF? All of these different operational definitions make it hard to understand structural commonalities and differences and compare results among different groups.

2. Why invent a new and obscure nomenclature, like OA* and OB* when there are already terms for at least OB*. Chris Dobson called them "prefibrillar oligomers" because they are transient species that occur kinetically early before fibrils evolve. The originators of the A11 antibody show that A11 is specific for "prefibrillar oligomers". I know that Professor Dobson is not able to address this, but he is an author.

3. Results: What is the N-terminal region of aS specifically?

4. Is the molarity of SF and LF based on monomer or particle concentration?

5. "We then analyzed the abilities of the OB*, SF and LF species, at a concentration of 10 μ M monomer equivalents, to interact and bind SUVs with the same lipid composition described above" It isn't clear what the rationale is for this.

6. The intracellular ROS burst is one of the inflammatory sequelae associated with amyloid internalization (Currais, A., Fischer, W., Maher, P., and Schubert, D. (2017) Intraneuronal protein aggregation as a trigger for inflammation and neurodegeneration in the aging brain, *FASEB J* 31, 5-10.)

7. The title of the figure legend for Figure 2 says that surface localization of aS fibrils does not correlate with toxicity, but the data show ROS instead.

8. Fig. 4D, E. The intracellular A11 signal could also be due to the conversion of endogenous aS to A11 + species. It might be interesting to do a titration of aS A11+ oligomers added to see if the internal fluorescence can exceed the total added fluorescence.

9. General: The term internalized is used loosely throughout to mean the protein is intracellular. The term internalize is generally understood in cell biology to mean endocytosis via invagination of a plasma membrane vesicle. The other mechanism is that some peptides and proteins "penetrate" the membrane, passing through the bilayer directly into the cytosol. This causes confusion. The authors do not experimentally distinguish these mechanisms, but it they would not be expected to be found in the nucleus if they were in endocytic vesicles. This distinction is an important one because the endocytosed molecules generally remain in the lumen of the vesicle and get degraded, unless an inflammatory reaction causes the leakage of vesicle contents into the cytosol. Things that penetrate the bilayer often cause leakage or pore formation in the plasma membrane.

10. The title should be more descriptive of what the novel mechanism is if you want to catch the attention of the reader.

Reviewer #1 (Remarks to the Author):

This is an interesting study detailing a mechanism of cell toxicity, i.e. the release of toxic oligomers from alpha-synuclein (a-syn) fibrils bound to the cell membrane, of likely relevance to disease pathogenetic processes.

We thank the reviewer for her/his positive comments on our work.

Important issues should be addressed, however, to strengthen interpretation of the results and legibility of the paper. Quite importantly, some of the conclusions of the manuscript do not seem to be in line with the results presented and should therefore be reformulated or better justified.

1) The last sentence of the abstract needs to be clarified. The sentence indicates that a-syn fibrils are responsible for neuron-to-neuron spreading of a-syn pathology. However, cell-to-cell passage of a-syn is not investigated as part of this current study and in fact, as further indicated below, the present findings do not really support the likelihood that fibrils, if released in the extracellular space, may be easily taken up by other cells.

The Reviewer is correct. Our intention was to highlight the additional, previously proposed toxic mechanism of fibril spreading, which could be complementary to that we have reported here and, in fact, they could both be involved in the global mechanism of fibril toxicity. However, as the reviewer has pointed out, our study does not provide information of the fibril cell-to-cell spreading mechanism, so the last sentence of our previous Abstract, as it was written before, needed to be corrected to account for this fact. We have, therefore, re-written this statement as follows: "In addition to previous evidence that α S fibrils can spread in different brain areas, our results reveal that α S fibrils can also release oligomeric species responsible for an immediate dysfunction of the neurons in the vicinity of these species".

2) The initial sets of experiments were performed using synthetic membranes. From these experiments, it is concluded that (i) the beta-sheet cores of short fibrils (SF) and long fibrils (LF) do not interact significantly with lipid bilayer, (ii) toxic oligomeric species (OB*), SF and LF are all able to interact with synthetic vesicles, and (iii) OB* is particularly effective in disrupting membrane integrity. Taken together, these findings probably indicate that fibrils may interact with the membrane surface but do not cause significant disruption of the lipid bilayer. This conclusion, if correct, should be clearly stated.

The Reviewer is correct. In order to summarise the results of the initial set of experiments, we added this statement "Taken together, these findings indicate that fibrils can interact with the membrane surface likely through the N-terminal region of the protein but do not cause significant disruption of the lipid bilayers as they cannot insert their β -sheet core, whereas prefibrillar OB* are particularly effective in disrupting membrane integrity by inserting their partially-formed β -sheet core" at the end of the first sub-section of Results on page 6.

3) In general, the description of the results should be made clearer. While reading the paper, data description in the text cannot always be easily correlated to the corresponding figures. It also appears that reference to specific figures may be erroneous. For example, reference is made to Figure 2C (instead of Figure 2B) after the statement "70%, 12% and 6% of the OB*, SF and LF samples were internalized...." (page 6).

We checked extensively the manuscript to make sure that references to all figures were correct.

4) Data in Figures 2B, C and D are reported as percent of endogenous a-syn. Did endogenous a-syn remain stable or did it change after treatment with OB*, SF or LF?

Under our experimental conditions, SH-SY5Y untreated cells showed very low and negligible levels of endogenous α S (green channel). We included a representative confocal image of untreated cells in the new figure 2A of the revised manuscript. However, all percentages were calculated with respect to endogenous α S in untreated cells. If endogenous α S levels change upon OB*/SF/LF treatment, this will be minor and will not affect the calculations. Nor is endogenous α S recruited to the intracellular aggregates as described later in the text (second last sub-section of Results on Pages 12-13).

5) Data on the internalization of OB*, SF and LF are reported after incubations for 1h (page 6). Do features of internalization of OB*, SF or LF change after longer incubations?

To clarify this point, OB*, SF and LF internalization was further explored following cell incubations for 0, 1, 3, 6, 24 h by using immunostaining. The new images of this time course analysis and the corresponding kinetic plots are shown in the new figure S3C and described in the Results section on page 7.

6) The use of N-acetylated OB* should be explained/justified.

The use of N-acetylated OB* has been justified on page 9, as requested by the Reviewer.

7) "Taken together our results indicate that a-syn short fibrils..." (page 8). This is an important summary of the results on internalization and toxicity of a-syn fibrils. Why no reference is made in this summary to the results obtained with long fibrils? Were the results obtained with SF and LF overall consistent?

The reviewer is correct. We had overlooked any description of the long fibrils. According to this recommendation, the main results obtained with long fibrils (LF) were included in the last paragraph of the sub-section indicated by the reviewer (page 9) to point out this relevant data. In addition, we have also added a description on LF before, when describing the Ca^{2+} influx (page 8).

8) A critical set of experiments in support of the main conclusion of this paper (i.e. that a-syn fibrils release oligomers that are ultimately responsible for their toxicity) is carried out with the conformation-sensitive antibody A11. Other conformation-sensitive antibodies are available and, given the importance of these experiments, validation of some of the results obtained with A11 using at least one more conformation-sensitive antibody is warranted.

Following the referee's suggestion, the release of toxic oligomers from fibrils was confirmed by a new immunostaining analysis using the conformational-sensitive polyclonal Syn33 antibody (Sigma Alsdreich), that recognizes the oligomeric species of α S with a good specificity as assessed by a dot-blot assay (figure S8D). The new data are described in the Results section on pages 11-12 and in Figures S8E,F.

9) The time points at which analyses were made should be justified and, if needed, made more consistent. For example, visualization of fibrils and oligomers using STED microscopy was carried out after incubations for 14 h. Why was this time point chosen whereas, for other experiments, analyses were made after 24h?

Following the reviewer's criticism, we investigated cells treated for 24 h with OB* and SF using STED microscopy. In addition, we have quantified the green intracellular and extracellular

fluorescent signals. The new data are shown in the new figure 5B,C and described in the Results section on pages 12-13. We had initially assumed that the intracellular levels of α S in cells treated for 14 h were similar to 24 h, according to the results shown in the time-course analysis (figure 4D,E).

10) An important control experiment involved the addition of monomers to SF or LF in order to evaluate if formation of oligomers in the extra-cellular space would result in their internalization and ultimately their toxicity (in the form of calcium influx). Cells were incubated with SF or LF in the presence or absence of monomers for 15 min or 180 min. These incubation times may not be sufficient for the generation of oligomers, their internalization and the production of toxic effects. Please explain/justify.

The reviewer correctly argues that 180 min may not be sufficient for the generation of oligomers from monomers M by a seeding process in the presence of SF or LF, ultimately resulting in calcium influx in the cells. We have chosen a time of 180 min, as the toxicity observed with cells exposed to SF/LF was already evident within 180 min (Figure 3C). This point was not clear in our text, thus we have added this explanation on page 13.

11) Images shown in Figure 6A are not very clear in representing the effect of A11 or OC on the internalization of a-syn after incubations with SF and, particularly, LF.

The Reviewer is correct. In order to show the results of figure 6A with higher quality, we have improved the resolution of the green signal and added some arrows to help the identification of the intracellular small green dots. In addition, we clarified the statement in the Results section on pages 13-14.

12) The ability of A11 to block a-syn internalization after exposure to SF and LF is an important finding. Is this finding due to the fact that this antibody binds oligomeric species that are first released from SF in the extracellular space and then taken up by cells? Please explain/interpret.

The Reviewer is correct. To better clarify this important issue, we added this sentence " This data suggests that A11 antibody binds oligomeric species that are released from SF/LF, thus preventing their cellular uptake and their downstream effects" on page 14.

13) The conclusion that fibrillar a-syn is responsible for neuron-to-neuron spreading of a-syn (page 14) is not supported by any of the data shown in this paper. In fact, findings of this study obtained with SH-SY5Y cells indicate that only 10% or less of fibrils added to the culture medium were actually internalized; most of these fibrils remained bound to the membrane but were not taken up (page 6). Data in Figure 4C also suggest that, with SF, very little or no significant amounts of fibrils gained access into the cytosolic space. Finally, data in Figure 5 clearly show lack of internalization of a-syn fibrils. How would these data be compatible with neuron-to-neuron transfer of a-syn fibrils? These data seem inconsistent with previously published findings showing uptake of a-syn fibrils in vitro (please explain). Overall, the present results strongly support a key role of toxic oligomers in pathogenetic processes (please comment). Is it possible that these oligomers, once generated and internalized from fibrils bound to the cell membrane, could also play an important role in neuron-to-neuron transfer of a-syn pathology?

We agree with the reviewer that none of our data present in this manuscript support the conclusion that fibrillar α S is responsible for neuron-to-neuron spreading of α S pathology. This conclusion comes from other papers, mainly contributed by other authors, which are now duly referenced in our revised text (Page 16). Our own results suggest that α S fibrils can contribute to pathogenesis by releasing oligomers causing an immediate dysfunction of the neurons. These mechanisms are not mutually exclusive or in contradiction, in our opinion. First, some α S fibril polymorphs (probably different from our SF/LF species) have been observed to be

internalized by cells (Volpicelli-Daley et al., 2011; Brahic et al., 2016) and possibly contribute to pathology spreading (Desplats et al. 2009; Luk et al., 2012; Masuda-Suzukake et al., 2013; Peelaerts et al., 2015; Prusiner et al., 2015; Froula et al. 2019) and neurodegeneration has to be envisaged within a scenario of fibril polymorphism. Second, oligomeric species of the type studied by us could also contribute to pathogenesis by generating new fibrils, following their neuronal internalization, therefore contributing to neuron-to-neuron spreading of α S pathology with a transfer following the release of oligomers from fibrils and their subsequent internalization. We have expanded our Discussion on page 16 to explain these points.

Reviewer #2 (Remarks to the Author):

General Comments:

In this manuscript, the authors define the nature of α -synuclein species that induce neuronal damage, and in particular describe a mechanism by which short fibrils may release oligomeric species that induce neuronal dysfunction. Overall this is interesting to read, and demonstrates a new potential phenomenon that would be important to the field. The questions raised below are both conceptual for the selection of model and approach to characterise the species inside cells, and technical with regards to the experimental paradigms.

1) Many of the results shown here have been already demonstrated by this group, and others, in a range of cell systems and in vivo models: namely that the oligomeric species either recombinant OB, or endogenously generated, can exert toxicity, through induction of ROS and oxidative stress, membrane perturbation, and cell death. The demonstration of some of these well-established features of synucleinopathy mostly in a neuroblastoma cell line using recombinant species, and the significant new mechanistic advances, over the published phenotypes, was not clearly described.

According to the Reviewer's suggestion, we have rephrased some of the key statements in Abstract, end of Introduction, Discussion and Conclusions.

2) The relevance of this model to the aggregation of alpha-synuclein, and the choice for this approach over existing transgenic animal or human models is not clearly articulated.

As human dopaminergic neurons, the cells mainly affected in PD, are difficult to obtain and maintain as primary cells, current PD research is mostly performed with permanently established neuronal cell models, in particular the neuroblastoma SH-SY5Y lineage. This cell line is frequently chosen because of its human origin, catecholaminergic neuronal properties, ease of maintenance. In spite of the large use of this cell line, we have also validated the release of oligomers from α S fibrils and the mechanism of membrane perturbation by employing primary rat cortical neurons. More importantly, after the reviewer's suggestion, in the revised version of the manuscript we have also included additional experiments on human iPSC-derived dopaminergic neurons to address our main results in a more relevant model of PD (see below our response to points 3,9,12).

3) The kinetics of the internalisation, the kinetics of aggregation and disaggregation and oligomer release, and the mechanisms of induction of ROS and membrane perturbation, and even toxicity in a neuroblastoma cell line are likely to differ significantly from human neurons, and therefore will the results here be relevant for human brain and disease: it would be improve significantly if these effects were present in other in vitro or in vivo models.

Following the reviewer's suggestion, we have extended our analysis to cellular models other than neuroblastoma SH-SY5Y cells. In particular, we have followed the internalisation of A11-

positive α S species in iPSC-derived dopaminergic neurons (Figure 4C, description on pages 10-11), we have carried out a new analysis of caspase-3 activation in iPSC-derived dopaminergic neurons (Figure 3D,E, description on page 9), calcein release using rat primary cortical neurons (Figure 3A, description on page 8), Ca^{2+} uptake using the same neurons (Figure 3B, description on pages 8-9) and MTT reduction using the same neurons (Figure 3F, description on page 9). We have not carried out extensive kinetics, given the difficulties and limitations of these cell lines, but all the data at representative time points confirm those obtained with neuroblastoma cell lines at corresponding times. The reviewer also mentions "kinetics of aggregation", but we have not followed time courses of aggregation in this study, either in vitro or in cell lines.

4) The finding of the presence of the toxic oligomeric species on addition of SFs is the most novel finding of this study, and very interesting to the field. The data for this rests on observations made by quantifying fluorescence from immunocytochemistry utilising antibody-based tools that are difficult to be certain of their specificity for structural species within cells. Orthogonal biophysical approaches in cells that can prove this hypothesis would greatly add to this work.

Following the reviewer's suggestion, we performed a new STED analysis in cells treated for 24 h with OB*/SF fluorescently labelled with AF488 (AF488-OB* and AF488-SF). This approach is immunocytochemistry-independent and antibody-independent, as requested by the reviewer. As explained in Supplementary Materials and Methods, the fluorescently labelled α S molecules carrying the AF488 dye (Invitrogen, Carlsbad, CA, USA) were obtained by using the N122C mutational variant of alpha-synuclein, allowing the dye molecule to react with the thiol moiety of Cys122. The labelled protein was then purified from the excess of free dye by chromatography. Fluorescent OB* and SF were generated by mixing 90% and 10% of unlabelled and labelled α S, respectively. The low ratio of labelled to unlabelled monomers and the C-terminal position of Cys122, which is not involved in the aggregation process and membrane interaction, ensured the absence of significant modifications to the properties of the oligomers and fibrils. The new data are showed in the new figure 5E,F and described in the Results section on page 13. In addition, the release of oligomers from SF was also showed in new experiments carried out on cell-free samples using DLS (Figure S8B and description on page 10).

5) The authors provide a semi-quantitative assessment and correlation of different structural forms and their effects in cells. Their comparison of one species vs another is based on the number of monomer equivalents in any aggregate. Yet in the discussion they speculate that the toxic aspect is due to the ends of any fibril, suggesting that the number of ends is the relevant parameter for toxicity. In that case, they would need to make equivalent the concentration of fibrils (numbers/volume) rather than the monomer equivalents, otherwise it is not possible to infer results between SF and LF (in which the LFs would be fewer in number as more monomer equivalents, and therefore fewer open ends).

We agree with the reviewer's argument. Thus, we performed an estimation of the number of monomers per fibril for SF and LF (see below). We then compared the toxic effects (ROS generation and Ca^{2+} uptake) of equivalent concentration of SF and LF (numbers/volume) in SH-SY5Y cells following aggregate treatment. We have expanded our Results on page 12 (main text) and a new paragraph in the Supplementary Materials and Methods on page 12 to explain this point.

We have estimated the same fibril number concentrations for SF and LF (corresponding to a higher mass concentration for LF). For the OB* oligomers we took the average number of monomers per oligomer from AUC analysis (Chen et al. PNAS 2015). This was ca. 30 monomers. For the fibrils we can estimate the number of monomers per fibril using the following equation:

$$N = \frac{\rho V N_A}{MW} = \frac{\rho \pi \left(\frac{h}{2}\right)^2 l N_A}{MW}$$

where N is the number of monomers per aggregate, N_A is the Avogadro's constant, MW is the molecular weight of the monomeric components, ρ is the density of the aggregate (1.35 g/cm³), V is the volume of the fibril, and h and l are the height and length of the fibril, respectively (derived from AFM). The numbers obtained are ca. 60 monomers for SFs and ca. 500 monomers for LFs.

Thus, 0.3 μ M concentration (monomer equivalents) of SF and 3 μ M concentration (monomer equivalents) of LF give rise to ca. 0.005 μ M particle concentration in both samples. We have therefore compared 0.03 μ M SF with 0.3 μ M LF. Our results of ROS and Ca²⁺ showed a similar toxic effect of both fibrils. The new data are shown in the new figure S8G and described on page 12.

6) There is a lack of description of the methods used to characterise the samples, and key details such as the purity of the samples for the species of interest. Whilst this is referenced, the nature of the species and their heterogeneity and variability is key here where biological effects are measured, and should be included. There is also a lack of description of the methods for the image analysis that derives % values – for example the image analysis pipeline that generates the values for intracellular, extracellular and total α -synuclein; and whether these are intensity or puncta values, and how the normalisation of this data occurred.

Samples of highly pure α S species were tested to be stable on the timescale of all the experiments reported in the present study. The assemblies were analyzed by atomic force microscopy (AFM, Supp. Figure 1A-E) to evaluate their morphology. The secondary structure content was determined by using far-ultraviolet circular dichroism (far-UV CD) and Fourier Transform Infrared (FT-IR spectroscopy (Supp. Figure 1G-H), together with the cross- β structure by X-ray diffraction (Supp. Figure 1I); the tinctorial properties were then evaluated by Thioflavine T (ThT) fluorescence (Supp. Figure 1J), and the solvent-exposed hydrophobicity by 8-anilino-1-naphthalenesulfonic acid (ANS) fluorescence (Supp Figure 1K). These results were reported in Figure S1 and described too succinctly in the text. We have therefore expanded the first sub-section of the Results section on page 4 to include this description.

Intracellular and extracellular α -synuclein-derived fluorescence was measured in about 70-80 cells, in three different experiments, using ImageJ software (National Institutes of Health, Bethesda, MD, USA), in regions of interest, whose dimensions remained constant for all the analyzed images. Intracellular α S fluorescence intensity was assessed in median planes by selecting multiple ROIs per cell in the cytoplasmic compartment, excluding the cellular membrane, after background subtraction. Similarly, the extracellular one was measured in basal, median and apical planes by selecting multiple ROIs per cell in the membrane compartment excluding the cytoplasmic compartment, in order to include only the signals arising from the species interacting with the outer leaflet of cell membranes. The fluorescence intensities were expressed as fractional changes above the resting baseline ($\Delta F/F$), where F (taken as 100%) represents the average baseline fluorescence of untreated cells. All these details are now reported in a new paragraph in the Method section of the Supplementary text on pages 11-12.

In addition, we have added as figure S3A a new quantification analysis of the α -synuclein-derived fluorescence reported in figure 2A by counting the number of intracellular puncta per cells using ImageJ software or the manual method. This quantification method generated data consistent with previous one also for the time course analysis reported in figure 4D.

Specific comments:

1) Characterisation of species. Why and how are the kinetics of calcein release accounted for the structure of the SF vs LF. How many aggregates are at the lipid membrane, and how transient is the disruption? What is mechanism of membrane disruption in OB vs SF vs LF? Is it that more aggregates insert or that for any one aggregate, it induces greater disruption? This would be relevant information for understanding the effect of OB vs SF vs LF in cells.

When describing the lower and slower release of calcein from SUVs caused by SF/LF relative to OB* in the Results section (Figure 1C), we have not yet described a sufficient amount of data to explain why and how the kinetics of calcein release with SF/LF are different from those of OB*. Indeed, the data of OB* release from fibril ends is described later. For this reason, we have a full description in the Discussion section on pages 15-16, explaining that SF and LF are less toxic and have a slower toxicity because they need time to release oligomers that, even after prolonged time, cannot be as many as those of a pure OB* sample. However, we did not mention in the Discussion section the observations of calcein release from SUVs. We have therefore expanded the Discussion section also including the following statement on page 15: "This also explains why fibrils caused a minor and slower calcein release from SUVs" to describe that the less intense and slower calcein release observed in SUVs with SF/LF is similar to that observed in cells and due to oligomer release by fibrils.

2) Interaction and internalised species: There is a lack of detail of this analysis: what is measured here as the morphology of the signal as well as intensity varies with species? (Intensity of fluorescence on ICC is not normally a reliable quantifiable variable). Why use the ratio of one plane to another in the cell, rather than analysing the fluorescence through the volume of a z stack? The control data for this analysis approach is not shown. This could be not possible in polarised cells such as neurons for example (and there is no clear evidence that a-syn is evenly distributed through the cell on uptake). What is the threshold set for endogenous a-syn? Is it the same for all samples and cells? How is intracellular alpha-syn measured vs extracellular?

As described above, the interaction and the internalization of α S species were assessed by quantifying the α S-derived green fluorescent signal of different optical sections (including basal, median and apical planes), in order to accurately discriminate the intracellular from the extracellular signals. This is described on page 6 of the revised manuscript. This estimation is not possible by simply measuring the fluorescence intensity through the total volume of a z-stack. Under our experimental conditions, SH-SY5Y untreated cells showed very low and negligible levels of endogenous α S (green channel) with respect to those observed in cells treated with OB*, SF or LF (see figure 2A). To point out this evidence, we included a representative confocal image of untreated cells in the new Figure 2A of the revised manuscript.

3) Why is there no monomer or OA uptake? Many papers have added monomers to cells and shown biological effects, and monomeric uptake is well documented.

Our semi-quantitative analysis of the confocal images of neuronal cells treated with either M or OA* indicates a weak interaction of these species with the cellular membranes and a small, albeit significant, increase in intracellular α S levels after 1 h with each a-syn species (figure 2A and S3A). Here we did not investigate M and OA* uptake at time of incubation longer than 1h, as in our experimental conditions neither M nor OA* could cause immediate or delayed neuronal dysfunction up to 24h (figure 3, S5, S6 and S7). It was out of the focus of our current work.

4) Fig 2 C,D: 'Bell shaped curve for intracellular and total therefore the amount of oligomers internalised into cells directly proportional to those interacting with membrane' How is this result inferred or measured? How did the authors determine the fluorescence signal arising from the plasma membrane?

As described above, membrane interaction and the internalization of α -synuclein species were assessed by quantifying the α -synuclein-derived green fluorescent signal of different optical sections (including basal, median and apical planes), in order to accurately discriminate the intracellular from the extracellular signals. This is described on page 6 of the revised manuscript. Intracellular and extracellular α S-derived fluorescence was measured as above reported (see point 6).

In the images there is minimal overlap of membrane and α -synuclein so do the authors presume that the traversing across by endocytosis is the %age that interacted with membrane? How did they decide this assumption is true – is this referenced?

To validate the accuracy of our data analysis of intracellular vs extracellular pools, we performed a new analysis using a mild protease digestion with 0.05% trypsin at 4 °C to remove the surface-bound OB*. In these conditions a marked amount of intracellular α S aggregates was evident at the median planes of SH-SY5Y cells. These results are described on page 6 and Figure S3B.

Moreover, the inhibition of the endocytic pathway, either by a pharmacological treatment with dynasore or by a temperature-mediated blockage of active processes, significantly decreased the amount of intracellular aggregates in treated cells, indicating that the influx of OB* through the membrane bilayer cannot be attributed only to a passive diffusion mechanism, but also to endocytosis. These results are described on page 6 and Figure S3B.

5) A correlation of ROS using live imaging, and intracellular protein level by ICC is clearly demonstrated. Proof of this would be to measure the amount of internalised α -synuclein in the same experiment as the ROS measurement for confirmation of the correlation.

According to the reviewer's suggestion, we performed a new confocal analysis of living SH-SY5Y cells treated for 24 h with fluorescently labelled aggregates (AF488-OB* and AF488-SF) *in vivo*, monitoring concomitantly the build-up of mitochondrial superoxide and the intracellular α S levels in cells not subjected to a fixing step. The new data are showed in the new figure 2C,D and described in the Results section on page 8.

6) Calcein leakage from SHSY5Y cells, or calcium influx experiments: What does a basal calcein reduction mean biologically? Is the membrane disruption constant and aggregates are remaining in the membrane? (This is not apparent on confocal imaging.) or do the aggregates pass through membranes and leave the disruption stably?

Calcein acetoxymethyl (Calcein-AM) is a substrate that passively crosses the cell membrane and in the cytosol is hydrolyzed by the enzyme esterase to a polar green-fluorescent product (calcein) that is retained into cells with intact membrane (Papadopoulos et al., 1994). The fluorophore calcein displays good retention characteristics (several hours after fixation) and low pH sensitivity. This method analyses cell membrane integrity. Calcein leakage suggests a permanent disruption of neuronal bilayers following the aggregates penetration through membranes and their intracellular localization. The meaning of the calcein release test is described and duly referenced on page 8.

7) Calcium influx should be compared to the same cell at basal level rather than 'untreated cells', as otherwise there is no control for the loading of the dye which is different in different cells

Calcium influx in neurons is transient and spikes related to transient binding of aggregates to membranes, therefore how does a snapshot at a fixed timepoint capture this in the rodent neurons. Calcium imaging in neurons would require dynamic imaging and an analysis of the number of spikes and their amplitude, unless there is clearly a change in basal fluorophore signal and a basis for this?

According to the reviewer's suggestion, we performed a new *in vivo* dynamic Ca^{2+} analysis of SH-SY5Y cells upon 30 min of treatment with OB* and SF. The new data are shown in the new figure S6C and described in the Results section on page 9.

8) What are the kinetics of calcium influx for SHSY5Y cells: is it the same as for calcein reduction? Where is the control in calcium free media?

The kinetics of Ca^{2+} influx are reported in Figure 3C, and appeared to be exponential for all the analysed species, with apparent rate constants of $1.1(\pm 0.1) \cdot 10^{-3} \text{ s}^{-1}$ for OB*, $0.8(\pm 0.2) \cdot 10^{-3} \text{ s}^{-1}$ for SF and $0.4(\pm 0.1) \cdot 10^{-3} \text{ s}^{-1}$ for LF. The control in Ca^{2+} -free medium can be found in the new figure S6A, where SH-SY5Y cells were treated for 15 min with OB* at 0,3 μM (monomer equivalents) in the absence of calcium. In this condition we did not observe any significant increase in Ca^{2+} entry, indicating that the influx of Ca^{2+} ions was exclusively from the extracellular medium and it was not released from the intracellular stores. We have also analysed the release of calcein in SH-SY5Y cells treated with OB* at 0.3 μM (monomer equivalents).

The results obtained showed an exponential decay in time, with an apparent rate constant of $5.8(\pm 2.1) \cdot 10^{-4} \text{ s}^{-1}$. The rate of Ca^{2+} entry is more rapid than calcein release, as observed here and for other oligomer systems (Zampagni et al. 2011). Ca^{2+} ions are smaller and can diffuse through small holes in the membrane. In addition, oligomer-induced activation of NMDA/AMPA receptors also cause Ca^{2+} ions to flow through the oligomer-affected membrane. By contrast, calcein is bigger and more extended perturbations are needed for this molecule to exit from the cells.

9) The MTT assay is a crude measure of viability, based on mitochondrial metabolism but not a specific indicator of mitochondrial dysfunction, which appears overstated. Why isn't the caspase 3 assay performed in neurons? Where is proof of membrane disruption or ROS being actually toxic to neurons where the mechanism is relevant? And how do these fairly different effects in cells interact to cause toxicity?

Following the reviewer's suggestion, we carried out a new caspase-3 analysis on human iPSC-derived dopaminergic neurons treated with OB*, SF and LF or untreated. The new data are shown in the new figure 3D,E and described in the Results section on page 9. In addition, membrane disruption was also probed on rat primary cortical neurons measuring both calcein release and Ca^{2+} influx (Figures 3A,B, description on pages 8-9).

10) Results 4: Characterisation of the A11 antibody with dot blot analysis looks very promising, but does not inform what A11 detects in cell lysates, or in ICC? Does it recognise other oligomers or aggregates in cells, and can that data be generated?

Following the reviewer's suggestion, we added a new image of untreated cells labelled with A11 in the new figure S8C, which showed the absence of any significant staining, and explained the significance of A11 antibody in the text on page 10. Moreover, the release of toxic oligomers from fibrils was confirmed by immunostaining analysis using another conformational-sensitive polyclonal Syn33 antibody (Sigma) (figure S8E,F), that recognizes the

oligomeric species of α S with a good specificity as assessed by a dot-blot assay (figure S8D). The new data are described in the Results section on pages 11-12.

11) Fig 4B – how is there no background in cells using a total α -synuclein antibody – is this adjusted in the images, or is it true finding?

This experiment is without cells and is now better explained in the text with a new paragraph section on pages 9-10.

12) Did the authors detect the presence of oligomers on adding SF and LF to neurons as well as neuroblastoma cells?

Following the reviewer's suggestions, we have carried out a new immunostaining analysis with A11 antibody in human iPSC-derived dopaminergic neurons treated with OB*, SF and LF for 24 h or untreated. The new data, are very similar to previous evidence in SH-SY5Y cells (new figures 4C and S8C) and are described in the Results section on page 10-11.

13) Why not perform this experiment in the absence of cells to demonstrate that oligomer release occurs without cells? In that case the dot blot characterisation of the A11 is relevant and the kinetics of release of oligomers can be determined far more accurately.

The experiments showed in the old Figure 4B had been performed in the absence of cells using the conformation-sensitive A11 antibody and the conformation-insensitive 211 antibody. We have extended the analysis to LF (Figure S8A) and to a larger number of time points for OB* and SF (0, 1, 3 and 24 h at 37° C) (Figure 4B). We explain better this point on page 10. In addition, we also performed a new dynamic light scattering (DLS) analysis of SF and OB* following different lengths of time. The new data indicate that the release of oligomers from fibrils can also occur *in vitro*, as reported in the new figure S8B and described on page 10.

14) If this does not occur in vitro in cell free systems, then what is the cellular mechanism that causes this?

It appears to occur in cell free system (See above). This is now clearly explained in the text on page 10.

15) Results 5: STED imaging – it was not clear why the authors are not using the A11 antibody to measure oligomers in these experiments? There is only qualitative analysis here, why no quantification of this data, or presentation of the precision and resolution of the imaging.

We used the mouse monoclonal 211 anti- α S antibody because it was raised against residues 121-125 of human α S (sequence DNEAY), but is unable to interact with endogenous rat α S (sequence SSEAY), therefore representing a good probe to monitor only exogenous (externally added) human α S species and not endogenous α S in rat primary neurons. This is explained on page 12. The A11 antibody was used in an independent analysis in figure 4. To address this point of the reviewer, we have quantified the 211-derived green intracellular and extracellular fluorescent signals of the STED experiments. The new data are showed in the new figure 5C and described in the Results section on pages 12-13.

16) The images with scale bar show that the fluorescent spots are roughly 500nm in size, much larger than diffraction limit of light and therefore not at super resolution. In these conditions, how does a spherical aggregate resemble an oligomer or a circular group of fibrils??

The review is correct. The resolution does not allow to visualise the correct size of an oligomer. The fluorescent labels suitable for STED analysis can generate different performance depending on experimental conditions, due to changes in signal intensity/stability (microenvironment, degree of labelling, mounting medium) and STED efficiency. In our experimental conditions we obtained a moderate STED performance. In particular, we can reveal intracellular small green dots of around 300 nm, or even less, in cells treated with both OB* and SF for 14 and 24 h. The smaller dots in the magnifications of Figure 5B are ca. 150 nm. In contrast, fibrillar α S aggregates outside the cells or attached to the membrane do not appear as green dots, but as elongated structures (high magnifications in Figure 5B). Thus, we can assume that the dots are oligomers and that the elongated aggregates are fibrils. It is unlikely that dots inside the cells represent circular groups of fibrils as they would appear with this size and shape outside the cells in samples treated with SF.

17) I did not follow how this therefore showed that the A11 fluorescent spots in previous experiments, and the 211 AB fluorescent signals both relate to exogenous a-syn (as the expts with A11 happen over 24hrs and so may easily recruit endogenous a-syn)

The experiments with the A11 antibody was used to monitor the type of aggregates inside the cells. It was a very useful tool as it detected OB*-like species but not M, OA*, SF, LF. It was central experiment to reveal that SF released oligomers and that these entered the cells. We agree with the reviewer that the A11-positive species present inside the cells might well include endogenous α S. To address this issue, we used the species-specific 211 antibody and STED microscopy in rat primary neurons to detect selectively exogenous human α S at high resolution. This conformation-insensitive antibody was raised against residues 121-125 of human α S (sequence DNEAY), but is unable to interact with endogenous rat α S, which displays a different corresponding epitope (sequence SSEAY). Accordingly, rat cortical neurons treated with the antibody in the absence of exogenous human α S did not reveal any green fluorescence arising from endogenous α S (Figure 5A). By contrast, rat neurons exposed to human α S-derived OB* or SF for 14 and 24 h exhibited high intracellular levels of 211-positive signal (green fluorescent punctae). These include only exogenous human α S. This point was clearly described in the manuscript but was not accurately summarized at the end of the description, so we understand that it might not have appeared clear to the reviewer. We therefore added the following statement on page 12: "These can only include exogenous human α S given the specificity of the 211 antibody".

18) If release of a-syn monomer by SF and secondary nucleation can occur, why not test this first in vitro rather than in cells?

The reviewer correctly argues that a more straightforward approach to reveal whether or not fibrils release monomers that subsequently convert into oligomers through secondary nucleation is to carry out experiments *in vitro* rather than in cells. Nevertheless, the aim of this experiment was not to assess if monomers can be released from fibril ends and converted into oligomers *per se*, but to assess whether the toxicity observed in cells with SF/LF could originate from oligomers formed from monomers in turn released from fibrils. For this reason, we have carried out the experiments in cells. Our measurable was toxicity in this case and we forced the presence of monomers by adding them directly to pre-existing fibrils. We admit that this point was not clear in our text and it has been reformulated on pages 13.

Results 6: Addition of the A11 antibody – if primarily recognises only OB, how does it prevent SF and LF internalisation by 85% (ie even more than OB internalisation). Prevention of internalisation of the species recognised by A11 (OB but also 85% SF and LF) abolished toxicity, in support of the previous experiments that prove that the OB is the toxic form.

The ability of A11 to block a-syn internalization after exposure to SF and LF is due to the fact that this antibody binds oligomeric species that are first released from SF in the extracellular

space and then taken up by cells. Thus SF/LF release toxic oligomers, but these are sequestered by A11 antibodies and their toxicity neutralized as shown previously (Kayed et al. 2003 Science). Again, this point was not clearly described and to better clarify this important issue, we add this sentence: "This data suggests that A11 antibody binds oligomeric species that are released from SF and LF on the membrane surface, thus preventing their cellular uptake and their downstream effects" on page 14.

The OC antibody is then used to show that prevention of internalisation of SF and LF, but not OB, abolishes toxicity. This is indirect evidence to support that SF and LF cause toxicity by oligomeric release, but how is the specificity of the OC antibody determined in cells?

The specificity of OC antibody was confirmed in cells by its negligible effect on the intracellular fluorescence and the toxicity caused by OB* (Figure 6A-C), acting as a negative control. This is explained on page 14. In this case OC antibody binds to SF/LF stabilizing them and preventing oligomer release.

Reviewer #3 (Remarks to the Author):

Summary:

This manuscript reports that a particular type of alpha-synuclein (aS) oligomer, OB*, displays a number of properties potentially related to toxicity and that short fibrils shed these oligomers from their ends. A lot of data is presented characterizing the oligomers and their properties, but the novel observation is the release of toxic oligomers from the ends of fibrils. While this is a novel and very interesting and significant idea of broad interest, the data on this and some of the other issues could be clearer and more convincing.

Major points:

1) Under the heading "aS fibrils gradually release oligomers that are ultimately responsible for their toxicity", the authors claim that short fibrils (SF), prepared by sonication of long fibrils (LF) release toxic oligomers. In Fig. 4B, they show a time dependent increase in A11 immunoreactivity for SF incubated in cell culture medium by confocal microscopy, but it isn't clear that this is a "release" of oligomers from SF or a conformation change in SF exposing the A11 epitope. Clearly, something is going on, but it isn't clear that it is release.

We agree with the reviewer and to address this issue we also performed a new dynamic light scattering (DLS) analysis of SF and OB* following different lengths of time. The new data indicate release of oligomers from fibrils, as detected by the appearance of small-sized species upon fibril incubation in PBS at 37 °C for variable lengths of time up to 24 h. These data are presented in the new figure S8B and described on page 10.

1.1) Similarly, the experiment in 4C is to incubate SF with cells and then perform western blotting on the membrane and cytosol fraction with the aS antibody 211. There is a difference for the SF treated samples between membrane and cytosol fraction in terms of intensity of staining being stronger in the membrane fraction and the pattern of bands is different between SF and OB, but I don't see any results that are consistent with the interpretation "release of oligomers". Key controls are missing, like what the preparations look like without incubation with cells and whether oligomers are "released in the absence of cells or SDS.

We agree with the Referee that WB analysis cannot be a definitive evidence and it is now being deleted. In addition, to confirm the release of oligomers from fibrils in the absence of cells or SDS, we performed a new dynamic light scattering (DLS) analysis of SF and OB* following

different lengths of time. The new data indicate that the release of oligomers from fibrils can occur also *in vitro* (the new figure S8B) described on page 10.

1.2) In order to demonstrate release, you would need to show separation of a smaller oligomer from its parent SF. If it is release from the ends, then the rate should be proportional to the number of ends, so you would expect LF to do it too but at a slower rate. LF were not examined in 4B and C. There should be some means of characterizing the size of the released oligomer, like sedimentation velocity or gel filtration. Additionally, a more detailed kinetic study of the release would be a significant improvement. Right now there is only one timepoint.

As described above, we performed a DLS analysis of SF and OB* following different lengths of time (new figure S8B and description on page 10). According to the reviewer suggestion, we also performed a new confocal microscopy analysis of SF and LF incubated in cell culture medium without cells at 0, 1 h, 3 h and 24 h at 37° C probed with 211 and A11 antibodies. The new data are shown in the new figure 4B and S8A and described on page 10.

More importantly, we performed an estimation of the number of monomers per fibril for SF and LF (see below). We then compared the toxic effects (ROS generation and Ca²⁺ uptake) of equivalent concentration of SF and LF (numbers/volume) in SH-SY5Y cells following aggregate treatment. We have expanded our Results on page 12 and a new paragraph in the Supplementary Materials and Methods on page 12 to explain this point.

We have estimated the same fibril number concentrations for SF and LF (corresponding to a higher mass concentration for LF). For the OB* oligomers we took the average number of monomers per oligomer from AUC analysis (Chen et al. PNAS 2015). This was ca. 30 monomers. For the fibrils we can estimate the number of monomers per fibril using the following equation:

$$N = \frac{\rho V N_A}{MW} = \frac{\rho \pi \left(\frac{h}{2}\right)^2 l N_A}{MW}$$

where N is the number of monomers per aggregate, NA is the Avogadro's constant, MW is the molecular weight of the monomeric components, p is the density of the aggregate (1.35 g/cm³), V is the volume of the fibril, and h and l are the height and length of the fibril, respectively (derived from AFM). The numbers obtained are ca. 60 monomers for SFs and ca. 500 monomers for LFs.

Thus, 0.3 μM concentration (monomer equivalents) of SF and 3 μM concentration (monomer equivalents) of LF give rise to ca. 0,005 μM particle concentration in both samples.

According to this estimation, we have chosen two different mass concentrations where the fibril concentrations are similar (in the left part of the bell-shaped graph because we have agglutination in the right) and we have compared 0.03 μM of SF with 0.3 μM of LF. Our results of ROS and Ca²⁺ showed a similar toxic effect of both fibrils. The new data are shown in the new figure S8G and described on page 12.

2) Another issue is whether the aS oligomers and fibrils have been internalized, where they are localized and how they are internalized. How do you know they are intracellular? How do you know they cross the membrane, as opposed to being endocytosed? Endocytosis can be blocked pharmacologically and by temperature.

The methods are not clear on these points, but it looks like the authors are just going by their subjective interpretation of images presented. This would not allow the localization to be determined with the precision needed to make their interpretation. It would be more convincing to show that the fraction that is presumed surface bound can be removed by washing at low pH or protease digestion. Alternatively, the aS could be colocalized with markers of different compartments in the endocytic pathway.

We have carried out the additional experiments requested by the reviewer (see below), but let us first describe the rationale of our previous analysis. The internalization of α S species were assessed by quantifying the α S-derived green fluorescent signal of different optical sections (including basal, median and apical planes), in order to accurately discriminate the intracellular from the extracellular signals. Intracellular α S fluorescence intensity was assessed in median planes by selecting multiple ROIs per cell in the cytoplasmic compartment, excluding the cellular membrane, after background subtraction. Similarly, the extracellular one was measured in basal, median and apical planes by selecting multiple ROIs per cell in the membrane compartment excluding the cytoplasmic compartment, to include only the signals arising from the species interacting with the outer leaflet of cell membranes. The fluorescence intensities were expressed as fractional changes above the resting baseline ($\Delta F/F$), where F (taken as 100%) represents the average baseline fluorescence of untreated cells. All these details are now reported in a new paragraph in the Methods section of the Supplementary text on page 12.

Furthermore, to probe whether OB* passively penetrated the cells or through the endocytic pathway, we performed a new immunostaining experiment by treating SH-SY5Y cells for 6 h at 37 °C with the endocytotic inhibitor dynasore, and at 4 °C to block endocytosis by a temperature decrease, as suggested by the reviewer. The new data are showed in the new figure S3B and described in the text on page 6.

To further investigate only the intracellular OB*, we performed a new immunostaining analysis by treating SH-SY5Y cells for 6 h at 37 °C and then adding 0.05% trypsin at 4 °C. This allows the detachment of all the proteins attached to the cellular membrane. The new data are showed in the new Figure S3B and described in the text on page 6.

2.1) TIRF microscopy would be more convincing of membrane penetration and the fact that the amyloid is crossing the membrane although it is not able to see deep within the cytosol.

We could not use TIRF microscopy due to the absence of this technique in our labs. However, to confirm our results, we performed a new STED analysis in cells treated for 24 h with fluorescently labelled aggregates (AF488-OB* and AF488-SF). As explained in Supplementary Materials and Methods, the fluorescently labelled α S molecules carrying the AF488 dye (Invitrogen, Carlsbad, CA, USA) were obtained by using the N122C mutational variant of alpha-synuclein, allowing the dye molecule to react with the thiol moiety of Cys122. The labelled protein was then purified from the excess of free dye by chromatography. Fluorescent OB* and SF were generated by mixing 90% and 10% of unlabelled and labelled α S, respectively. The low ratio of labelled to unlabelled monomers and the C-terminal position of Cys122, which is not involved in the aggregation process and membrane interaction, ensured the absence of significant modifications to the properties of the oligomers and fibrils. The new data are showed in the new figure 5E,F and described in the Results section on page 13.

2.2) It might be better to count puncta rather than the total fluorescence signal, since the interesting fraction is aggregated. There is quite a difference in terms of the number of particles that are observed in the nucleus and it would be easier to count this unambiguously (the nucleus is distinct from the "cytosol"). The significance of nuclear synuclein aggregates is not clear. I am not aware of any reports of them in human samples.

Following the reviewer's suggestion, we have performed a new quantification analysis of the α S-derived fluorescence reported in figure 2A and figure 4D,E by counting the number of puncta inside and outside the cells, using ImageJ software. The results obtained are in line with those acquired with the manual method and are reported in the new Figure S3A and described on page 6.

Minor points:

1) Introduction. There is considerable discussion about fibrils SF and LF and oligomers, but these terms are ill defined, both here and in the literature in general. What is the difference between an oligomer and fibril? What is the cut off for something to be called oligomeric or short fibril vs long fibril? Is there a structural definition that makes more sense than OA, OB, SF and LF? All of these different operational definitions make it hard to understand structural commonalities and differences and compare results among different groups.

The Reviewer is correct. In the first version of our manuscript we actually introduced the five species (M, OA*, OB*, SF, LF) without a clear definition. We reported a full characterisation of them in a supplementary figure and its legend (Fig. S1), but missed reporting key information in the main text. We have therefore maintained Figure S1 with a long legend (as it was before), but also added a 7-line description in the first paragraph of the Results section on page 4, to describe the key morphological, structural and tinctorial characteristics of the five species. The reader will be more oriented, we hope. Since the characterisation consists of experimental data, we believe it is more appropriate to define these characteristics at the beginning of the *Results* section rather than *Introduction*.

2) Why invent a new and obscure nomenclature, like OA* and OB* when there are already terms for at least OB*. Chris Dobson called them "prefibrillar oligomers" because they are transient species that occur kinetically early before fibrils evolve. The originators of the A11 antibody show that A11 is specific for "prefibrillar oligomers". I know that Professor Dobson is not able to address this, but he is an author.

It is important to differentiate between stable, kinetically trapped oligomeric forms, which we denote with the asterisk superscript, and the transient species formed during fibril formation or released from fibrils, without the asterisk superscript. Also we are maintaining the original nomenclature we first used to describe initially formed, mainly disordered oligomers (OA) and subsequently converted, partially β -sheet oligomers (OB) identified during fibril formation (Cremades et al 2012 Cell). Chris Dobson was still alive when we designed this study and participated to the first and second revision of this manuscript text, including the definition of the five species (M, OA*, OB*, SF, LF). Indeed, he insisted in differentiating kinetically trapped oligomeric forms and the transient species formed during fibril formation or released from fibrils, with and without the asterisk superscript, respectively. Moreover, there are at least three papers published before he passed away with these definitions of α S species (Cremades et al 2012 Cell; Chen et al. 2015 PNAS, Fusco et al. 2017 Science). We are confident that he would be happy with the definitions of the various species present in this manuscript. We are sure that Chris Dobson would also approve the definition of OB* as "prefibrillar oligomers", as suggested by the reviewer, and for this reason we have added the adjective "prefibrillar" in 10 different places, in the Abstract (page 2), Introduction (pages 3-4), first section of Results (page 4-6), Discussion (page 14) and Conclusions (page 16).

3) Results: What is the N-terminal region of α S specifically?

The N-terminal region of α S, consisting of residues 1-26, was found to be highly dynamic in the type-B* oligomers (Fusco et al. 2017 Science). We have added this numbering the first time we mention it in the Introduction (Page 3).

4) Is the molarity of SF and LF based on monomer or particle concentration?

Their comparison of one species vs another is based on the number of monomer equivalents in any aggregate. This is reported the first three times we mention molarity on pages 5-6, so that the reader becomes familiar with our molarity definition.

5) "We then analyzed the abilities of the OB*, SF and LF species, at a concentration of 10 μ M monomer equivalents, to interact and bind SUVs with the same lipid composition described above" It isn't clear what the rationale is for this.

The Reviewer correctly reports that the rationale for this experiment was not clearly explained. In order to study the interaction of the OB*, SF and LF species with synthetic vesicles of a physiologically relevant lipid composition, we used ssNMR and far-UV CD. With the first set of experiments we did not see interaction of SF or LF through their β -sheet core, in contrast to the OB*, however, the far-UV CD data indicated that the fibrils were indeed able to interact with the lipid vesicles, likely then with the free N-terminal region of the protein molecules in the fibrils, which are not part of the fibril core and can then interact with the lipid vesicles likely as in the monomeric protein. This explanation was added on page 5 of the revised manuscript. We hope it is now clear.

6) The intracellular ROS burst is one of the inflammatory sequelae associated with amyloid internalization (Currais, A., Fischer, W., Maher, P., and Schubert, D. (2017) Intraneuronal protein aggregation as a trigger for inflammation and neurodegeneration in the aging brain, *FASEB J* 31, 5-10.)

According to the reviewer's suggestion, we added the new reference in the Results section on page 7.

7) The title of the figure legend for Figure 2 says that surface localization of aS fibrils does not correlate with toxicity, but the data show ROS instead.

The Reviewer is correct. We have modified the title of the legend of the figure 2 to better explain that the extracellular fluorescence arising from the species bound to the cellular membrane does not correlate with their toxicity, whereas the intracellular one strongly correlates with the toxicity.

8) Fig. 4D, E. The intracellular A11 signal could also be due to the conversion of endogenous aS to A11 + species. It might be interesting to do a titration of aS A11+ oligomers added to see if the internal fluorescence can exceed the total added fluorescence.

According to the Reviewer's suggestion, we performed a new immunostaining analysis with A11 antibody of SH-SY5Y cells treated with increasing concentration of OB* and relative quantification of intracellular and total A11-derived fluorescence. In addition, we analyzed the intracellular and total A11-derived fluorescence following 15 min and 24 h of cell treatment with OB*. The new data are showed in the new figure S9 and described in the Results section on page 11.

9) General: The term internalized is used loosely throughout to mean the protein is intracellular. The term internalize is generally understood in cell biology to mean endocytosis via invagination of a plasma membrane vesicle. The other mechanism is that some peptides and proteins "penetrate" the membrane, passing through the bilayer directly into the cytosol. This causes confusion. The authors do not experimentally distinguish these mechanisms, but it they would not be expected to be found in the nucleus if they were in endocytic vesicles. This distinction is an important one because the endocytosed molecules generally remain in the lumen of the vesicle and get degraded, unless an inflammatory reaction causes the leakage of vesicle contents into the cytosol. Things that penetrate the bilayer often cause leakage or pore formation in the plasma membrane.

According to the Reviewer's suggestion, we have modified the term "internalization" with "penetration" or "uptake" throughout the manuscript.

10. The title should be more descriptive of what the novel mechanism is if you want to catch the attention of the reader.

We agree with the Reviewer. Thus, we have modified the title with "The release of toxic oligomers from α -synuclein fibrils induces neuronal dysfunction".

Reviewers' Comments:

Reviewer #1:

Remarks to the Author:

The Authors satisfactorily addressed the concerns I listed in my original review. The manuscript describes original data relevant to mechanisms of alpha-synuclein toxicity that could play a role in the development of human synucleinopathies. This revised version of the paper is significantly improved thanks to (i) additional data that more convincingly support the Authors' conclusions, (ii) more detailed description of experimental paradigms, and (iii) more thorough discussion of new and previously published findings.

Reviewer #2:

Remarks to the Author:

The authors have provided a detailed rebuttal and new data and I believe that this overall significantly improves the conceptual basis for the study, and clarifies some of the issues raised. The use of alternative approaches (other antibody and fluorescently labelled species) increases confidence in the findings. Overall, the study provides evidence that short fibrils release oligomers of the Type B structure that mediate the toxicity of short fibrils, a phenomenon that is observed both in the absence of cells, and with cells. This is a new and interesting observation to the field, and therefore will be of specific interest to the community, particularly with respect to the protein aggregation diseases. The potential impact of understanding specifically this mechanism endogenously *in vivo* is perhaps not articulated clearly for a broader readership.

The inclusion of new data and new experimental approaches has also raised some other issues of concern outlined below.

Major comments:

1. The revised version now incorporates results from 'iPSC derived dopaminergic neurons' as support that the mechanism is relevant in the human cells affected in disease. It is not clear from the methods or results what these cells are (progenitors or neurons), and there is no characterisation of their identity, their maturity or their function. It is also not clear what proportion of the culture is the cell type of interest (DAergic neuron), and it is well established that most protocols do not reach high levels of enrichment. Therefore, are the results from the cells of interest.

'Materials and Methods

Human iPSC-derived Dopaminergic Neuron Progenitors (Axol Bioscience, Cambridge, UK) were plated on 12-well plates containing glass coverslips, following the treatment with Poly-D-Lysine plus SureBond-XF Coating Solution. The cells were grown in differentiation medium at 37 °C with 5% CO₂ and the medium was refreshed every two days. On day 5 post-seeding, the differentiation medium was replaced with the maintenance medium, that was changed every two days. The experiments were performed between day 28 and day 32 of culturing.'

2. The authors describe the generation of mitochondrial superoxide correlates with the intracellular fluorescence of AF488-labelled a-syn species:

We also monitored concomitantly the generation of mitochondrial superoxide ions and intracellular aS levels in living SH-SY5Y cells exposed to 0.03, 0.1 and 0.3 μM of OB*/SF labeled with AF488 dye (Figure 2C). The increase in intracellular aS levels (green channel) correlated positively with the rise in mitochondrial superoxide production (red channel) in a dose-dependent relationship (Figure 2C,D). This result indicates that aS toxicity is associated with the ability of these species to penetrate the interior of the cell membrane, rather than binding on

their surface, and ultimately to enter into the cytosol.

If the purpose of this experiment is to correlate mitoSox and AF-488-syn then it is necessary to show the images from which the quantitative data is derived. The loading of the mitoSox probe into the mitochondria is notoriously variable, and often leads to a cytosolic distribution and reporting of cytosolic superoxide. What was measured here – the rate of superoxide production (slope of the intensity) or a static timepoint measurement? The methods/results do not show the positive control for the MitoSox imaging to confirm that it is indeed measuring mitochondrial superoxide. The conclusion of the experiment, namely that the fact that the mitoSox and a-syn correlate in intensity suggests that the a-syn toxicity is due to entry into the cytosol does not seem accurate here – the conclusion is that the generation of mitochondrial superoxide (if that is what is measured) is related to the level of intracellular a-syn.

3. Caspase 3 activity is measured in the iPSC dopaminergic neurons, and this assay is described in the methods simply as

Caspase-3 activity was then analyzed by the confocal scanning system described above, as previously reported (Casella et al., 2016)'

How is caspase 3 activation measured (is it ICC or an activity assay?) It is not clear from the methods or from the reference provided (which also cites a reference for the method)

4. Why was N acetylated aS introduced here – does it change the properties of OB, SF and LF, and how? There is a line stating that it does not but it is not clear that this is supported.

'Since aS is N-terminally acetylated in vivo (Anderson et al., 2006), representative experiments were repeated with OB* formed from the N-acetylated aS. Very similar results were obtained, using calcein release and MTT reduction as cell viability readouts (Figure S7A,B). Similar results were also obtained with the ultra-clean samples, where the recombinant protein was pretreated to remove potential lipopolysaccharides, probing ROS detection and MTT reduction (Figure S7C,D).'

Minor comments

1. Introduction

'Evidence is also accumulating that Lewy Body pathology can be transmitted from cell-to-cell in a spreading process in which different areas of the brain are slowly and inexorably affected (Desplats et al. 2009; Luk et al., 2012; Masuda-Suzukake et al., 2013; Peelaerts et al., 2015; Prusiner et al., 2015).'

The authors should consider the wording here – there is no evidence that the LB pathology itself 'transmits' from cell to cell...rather certain species transmit and induce seeding in neighbouring cells, so that they exhibit aggregation.

2. Results

'In brief, OA*/OB* are globular-like aggregates with heights of 5.1 ± 0.8 nm and 4.3 ± 0.9 nm, respectively; SF and LF have elongated morphologies with heights of 5.0 ± 2.6 nm and 6.2 ± 3.8 nm, respectively, and mean lengths of 57 nm and 520 nm, respectively. OA*/M show disordered secondary structure, while OB*/SF/LF have ca. 30-35%, 65% and 65% of their sequence in a β -sheet conformation, respectively. Unlike OA*/M, OB* have a significant but weak ThT binding, whereas SF/LF have a large ThT binding. Unlike OA*/M, OB*/SF/LF all have a strong ANS binding, indicating a high degree of solvent-exposed hydrophobicity.'

This is helpful characterisation of the properties of the species, and would be easier in a table for clarity and ease of reading.

3. Fig 1b: Misspelling of length on all axes

4. Fig 1c – why is the baseline of the OB so much higher than the SF/LF preps? Is the starting %release relevant- if so, then explain. If not, then normalize.

5. Results: 'Current PD research is mostly performed with immortalized neuronal cell models'

This is not an accurate statement, and it does not really give a good justification for choosing this line.

6. Results: 'The aS species (green channel) were counterstained and analyzed at the cellular apical, median and basal planes of the plasma membrane (red channel) parallel to the coverslip by confocal scanning microscopy'

There is no reference to this approach, or explanation how these planes are chosen and why.

7. 'Notably, when the surface-bound OB* were removed by a mild protease digestion with 0.05% trypsin at 4 °C, a large amount of intracellular aS aggregates was evident at the median planes of SH-SY5Y cells (Figure S3B), validating the accuracy of our analysis of intracellular vs extracellular aS pools.'

This could be clarified more – what does the trypsin digestion do to the different species, and to the plasma membrane? Why only measure in the median plane here?

8. Suppl fig 6 – 'in vivo' – throughout the manuscript the terms in vitro and in vivo are confusing – it may be better to use 'inside cells' and 'in the absence of cells' because neither is strictly in vivo.

9. 'Notably, we also observed faster kinetics for the rise of intracellular Ca²⁺ for OB* than SF and LF treatment at longer time of analysis in fixed SH-SY5Y cells (Figure 3C).'

How did they measure the kinetics in fixed cells?

10. Fig 4E – how does the A11 signal reach steady state if the SFs continue to release OB oligomers

11. Fig 5 E – how does the use of the AF488 affect the properties of the OB and the SF? Other papers are referenced here that utilize fluorescently labelled species, but does the use of the tag affect the release of oligomers?

Reviewer #3:

Remarks to the Author:

The authors have adequately addressed my comments

Reviewer #1 (Remarks to the Author):

The Authors satisfactorily addressed the concerns I listed in my original review. The manuscript describes original data relevant to mechanisms of alpha-synuclein toxicity that could play a role in the development of human synucleinopathies. This revised version of the paper is significantly improved thanks to (i) additional data that more convincingly support the Authors' conclusions, (ii) more detailed description of experimental paradigms, and (iii) more thorough discussion of new and previously published findings.

We thank the reviewer for her/his positive comments on our work.

Reviewer #2 (Remarks to the Author):

The authors have provided a detailed rebuttal and new data and I believe that this overall significantly improves the conceptual basis for the study, and clarifies some of the issues raised. The use of alternative approaches (other antibody and fluorescently labelled species) increases confidence in the findings. Overall, the study provides evidence that short fibrils release oligomers of the Type B structure that mediate the toxicity of short fibrils, a phenomenon that is observed both in the absence of cells, and with cells. This is a new and interesting observation to the field, and therefore will be of specific interest to the community, particularly with respect to the protein aggregation diseases. The potential impact of understanding specifically this mechanism endogenously in vivo is perhaps not articulated clearly for a broader readership.

According to the Reviewer's suggestion, we have rephrased several sentences in the manuscript and Supplementary text (see below).

The inclusion of new data and new experimental approaches has also raised some other issues of concern outlined below.

Major comments:

The revised version now incorporates results from 'iPSC derived dopaminergic neurons' as support that the mechanism is relevant in the human cells affected in disease. It is not clear from the methods or results what these cells are (progenitors or neurons), and there is no characterisation of their identity, their maturity or their function. It is also not clear what proportion of the culture is the cell type of interest (DAergic neuron), and it is well established that most protocols do not reach high levels of enrichment. Therefore, are the results from the cells of interest.

Materials and Methods

Human iPSC-derived Dopaminergic Neuron Progenitors (Axol Bioscience, Cambridge, UK) were plated on 12-well plates containing glass coverslips, following the treatment with Poly-D-Lysine plus SureBond-XF Coating Solution. The cells were grown in differentiation medium at 37 °C with 5% CO₂ and the medium was refreshed every two days. On day 5 post-seeding, the differentiation medium was replaced with the maintenance medium, that was changed every two days. The experiments were performed between day 28 and day 32 of culturing".

The Reviewer is correct. Since it was not clear that the analysis was performed on neurons rather than progenitors, we have now detailed the protocol in the revised Supplementary text on page 5. In addition, we have performed new experiments to characterize the identity and maturity of the differentiated neurons by analyzing the expression of tyrosine-hydroxylase (TH), a typical dopaminergic neuron marker, as well as the microtubule associated protein 2 (MAP-2). The analysis reveals that the dopaminergic neuronal population represents around 75 % of the cell culture on days 14-18. The new data are shown in the new Fig. 3D and described in the Results section on page 9. Since we found a high level of TH-positivity and

MAP-2 positivity in this time frame, we also showed the toxicity of α S fibrils (caspase-3 activation) and the release mechanism in such a relevant PD model on days 14-18.

2. The authors describe the generation of mitochondrial superoxide correlates with the intracellular fluorescence of AF488-labelled a-syn species:

“We also monitored concomitantly the generation of mitochondrial superoxide ions and intracellular aS levels in living SH-SY5Y cells exposed to 0.03, 0.1 and 0.3 μ M of OB*/SF labeled with AF488 dye (Figure 2C). The increase in intracellular aS levels (green channel) correlated positively with the rise in mitochondrial superoxide production (red channel) in a dose-dependent relationship (Figure 2C,D). This result indicates that aS toxicity is associated with the ability of these species to penetrate the interior of the cell membrane, rather than binding on their surface, and ultimately to enter into the cytosol”.

If the purpose of this experiment is to correlate mitoSox and AF-488-syn then it is necessary to show the images from which the quantitative data is derived. The loading of the mitoSox probe into the mitochondria is notoriously variable, and often leads to a cytosolic distribution and reporting of cytosolic superoxide. What was measured here – the rate of superoxide production (slope of the intensity) or a static timepoint measurement? The methods/results do not show the positive control for the MitoSox imaging to confirm that it is indeed measuring mitochondrial superoxide. The conclusion of the experiment, namely that the fact that the mitosox and a-syn correlate in intensity suggests that the a-syn toxicity is due to entry into the cytosol does not seem accurate here – the conclusion is that the generation of mitochondrial superoxide (if that is what is measured) is related to the level of intracellular a-syn.

According to the Reviewer’s suggestion, we have added the images of SH-SY5Y cells from which the quantitative data was derived in figure S5D, together with a positive control obtained by treating SH-SY5Y cells with H₂O₂ at 250 μ M. In this experiment we performed a static time-point measurement of mitochondrial superoxide production in living cells exposed for 1 h to increasing concentration of 488-labelled OB* and SF. This is now more clearly explained on page 7-8 of the revised Supplementary text. According to the Reviewer’s suggestion, we have also modified the last sentence of the paragraph on page 8 of the main text to improve accuracy.

3. Caspase 3 activity is measured in the iPSC dopaminergic neurons, and this assay is described in the methods simply as

“Caspase-3 activity was then analyzed by the confocal scanning system described above, as previously reported (Cascella et al., 2016)”

How is caspase 3 activation measured (is it ICC or an activity assay?) It is not clear from the methods or from the reference provided (which also cites a reference for the method).

Following the Reviewer’s suggestion, we have expanded the Methods section on page 9 of the revised Supplementary text by explaining the procedure for both iPSC-derived dopaminergic neurons and SH-SY5Y cells. In addition, we have explained in the same section that the FLICA reagent FAM-DEVD-FMK enters each cell and binds irreversibly and covalently to active caspase 3. The reagent bound covalently to the active enzymes is retained in the cell, while the unbound reagent diffuses outside and is washed away. The emitted fluorescence was then detected at 488 nm excitation by the confocal scanning system according to the manufacturer’s instructions (Immunochemistry Technologies, LLC, Bloomington, MN). Reference to Cascella et al. 2016 is avoided in this section.

4. Why was N acetylated aS introduced here – does it change the properties of OB, SF and LF, and how? There is a line stating that it does not but it is not clear that this is supported.

'Since α S is N-terminally acetylated in vivo (Anderson et al., 2006), representative experiments were repeated with OB* formed from the N-acetylated α S. Very similar results were obtained, using calcein release and MTT reduction as cell viability readouts (Figure S7A,B). Similar results were also obtained with the ultra-clean samples, where the recombinant protein was pre-treated to remove potential lipopolysaccharides, probing ROS detection and MTT reduction (Figure S7C,D).'

N-terminally acetylated α S was introduced to respond to another reviewer (reviewer 1 of our first re-submission). According to the latest comment of Reviewer 2, we added the following statement in the Results section on page 9 to support that the structural properties of α S species were not affected by the N-acetylation process: "It has been recently reported that no significant differences were observed in the structural properties of α S species following N-terminal acetylation (Fusco et al., 2017)".

Minor comments

1. Introduction

'Evidence is also accumulating that Lewy Body pathology can be transmitted from cell-to-cell in a spreading process in which different areas of the brain are slowly and inexorably affected (Desplats et al. 2009; Luk et al., 2012; Masuda-Suzukake et al., 2013; Peelaerts et al., 2015; Prusiner et al., 2015).'

The authors should consider the wording here – there is no evidence that the LB pathology itself 'transmits' from cell to cell...rather certain species transmit and induce seeding in neighbouring cells, so that they exhibit aggregation.

We have now rephrased the sentence in the Introduction section on page 3.

2. Results

'In brief, OA*/OB* are globular-like aggregates with heights of 5.1 ± 0.8 nm and 4.3 ± 0.9 nm, respectively; SF and LF have elongated morphologies with heights of 5.0 ± 2.6 nm and 6.2 ± 3.8 nm, respectively, and mean lengths of 57 nm and 520 nm, respectively. OA*/M show disordered secondary structure, while OB*/SF/LF have ca. 30- 35%, 65% and 65% of their sequence in a β -sheet conformation, respectively. Unlike OA*/M, OB* have a significant but weak ThT binding, whereas SF/LF have a large ThT binding. Unlike OA*/M, OB*/SF/LF all have a strong ANS binding, indicating a high degree of solvent-exposed hydrophobicity.'

This is helpful characterisation of the properties of the species, and would be easier in a table for clarity and ease of reading.

According to the Reviewer's suggestion, we have added a Table reporting the main biophysical and structural properties of the analyzed α S species. The new Table is shown in the revised Supplementary text on page 14.

3. Fig 1b: Misspelling of length on all axes.

We have corrected the misspelled word on all axes.

4. Fig 1c – why is the baseline of the OB so much higher than the SF/LF preps? Is the starting % release relevant- if so, then explain. If not, then normalize.

The % calcein release was determined as explained in the Supplementary Information, on page 3. Briefly, the fluorescence signal obtained at each time point for each protein species sample was compared with the fluorescence signal of the samples containing only SUVs (basal fluorescence) and that of the samples containing SUVs that have suffered a complete disruption after treatment with 1% v/v Triton X-100 (maximum fluorescence signal, i.e. 100% calcein release). According to the Reviewer's suggestion, we have rephrased the sentence in the Results section on page 5-6: "We found that OB* induced a rapid and substantial calcein release, with ca. 30 % of maximum release at the first time point of measurement, and reaching the maximum level (ca. 90% of maximum release) after 2 min of incubation at a protein:lipid ratio

of 1:100. SF/LF caused, however, a much slower and very significantly reduced calcein release (ca. 30% release after 5 min) under identical conditions (Figure 1C)".

5. Results: 'Current PD research is mostly performed with immortalized neuronal cell models'
This is not an accurate statement, and it does not really give a good justification for choosing this line.

The Reviewer is correct and according to her/his suggestion, we have now rephrased the first sentence in the second paragraph of the Results section on page 6.

6. Results: 'The aS species (green channel) were counterstained and analyzed at the cellular apical, median and basal planes of the plasma membrane (red channel) parallel to the coverslip by confocal scanning microscopy'

There is no reference to this approach, or explanation how these planes are chosen and why.

In the revised version of the manuscript, concretely on page 7 of the Supplementary Information, we provide a more detailed explanation as to how these planes were chosen. The usefulness of the approach in order to discriminate between intracellular vs extracellular α S pools was more carefully described on page 6 of the revised manuscript; a reference on this topic was also included (Fusco et al., 2017).

7. 'Notably, when the surface-bound OB* were removed by a mild protease digestion with 0.05% trypsin at 4 °C, a large amount of intracellular aS aggregates was evident at the median planes of SH-SY5Y cells (Figure S3B), validating the accuracy of our analysis of intracellular vs extracellular aS pools.'

This could be clarified more – what does the trypsin digestion do to the different species, and to the plasma membrane? Why only measure in the median plane here?

According to the Reviewer's request, we added a new sentence to clarify the trypsin digestion experiment in the Results section on page 6: "Notably, a mild protease treatment with 0.05% trypsin at 4 °C, that digested all the protein molecules exposed on the cell membrane, including the surface-bound OB*, drastically reduced the green fluorescent signal at the apical planes (Figure S3B), without any significant modification of intracellular α S aggregates at the median planes of SH-SY5Y cells (Figure S3B). These results indicate that the signal that was previously observed at that cellular plane arose, indeed, from the internalized OB*". We also added apical planes in Figure S3B, as requested by the Referee.

8. Suppl fig 6 – 'in vivo' – throughout the manuscript the terms in vitro and in vivo are confusing – it may be better to use 'inside cells' and 'in the absence of cells' because neither is strictly in vivo.

We agree with the reviewer's point and we have avoided the use of the "in vivo" term in this context.

- Pag. 11: "Confocal images showed a remarkable A11-positive signal in neurons treated for 24 h with OB*, SF and LF (Figure 4C) with respect to untreated cells (Figure S8C), indicating the presence of α S oligomeric species similar to OB* **inside the cells** upon treatment with SF/LF";

- Pag. 12: "To evaluate further the nature of the α S species that were found **inside the cells**, we used the super-resolution stimulated emission depletion (STED) microscopy (Bigi et al., 2020) on rat primary cortical neurons and the mouse monoclonal 211 anti- α S antibody (Figure 5A-D)".

9. 'Notably, we also observed faster kinetics for the rise of intracellular Ca^{2+} for OB* than SF and LF treatment at longer time of analysis in fixed SH-SY5Y cells (Figure 3C).'

How did they measure the kinetics in fixed cells?

The Reviewer is correct. The term "fixed" was not appropriate to describe our experimental procedure. The levels of Ca^{2+} were actually measured in living cells instead of in fixed cells. We have now clarified this point on page 9 of the revised version of the manuscript.

10. Fig 4E – how does the A11 signal reach steady state if the SFs continue to release OB oligomers

As the reviewer is pointing out, SF and LF release OB-like oligomers and this process is favoured in the presence of cells. According to our cell internalization studies, the release process, under the conditions of study, is maximal during the first 12 h (Figures 4D, S8F) and, correspondingly, the maximal increase in A11-positive species inside the cells occurs during the same time scale. When analyzing the different cellular readouts associated with the internalization of OB-like oligomers, in all cases similar trends were observed, with an exponential or sigmoidal increase reaching a plateau after ca. 12-24 h of incubation.

11. Fig 5 E – how does the use of the AF488 affect the properties of the OB and the SF? Other papers are referenced here that utilize fluorescently labelled species, but does the use of the tag affect the release of oligomers?

According to the Reviewer's suggestion, we added a new sentence on page 13 to clarify that this experiment was performed with a low ratio of AF488 labeled OB* and SF (generated by mixing 90% and 10% of unlabeled and labeled α S monomers), thus ensuring the absence of significant modifications to the properties of OB* and SF as previously reported (Fusco et al., 2017). Moreover, the STED images indicated that the use of the tag doesn't affect substantially the release of oligomers (Fig 5E). In particular, human SH-SY5Y cells treated with OB* and SF labeled with the fluorescent AF488 dye clearly showed small oligomers with globular morphology on the membrane surface and inside the cells and only inside the cells, respectively (Figure 5E,F). These results are consistent with those obtained with OB*/SF species detected with A11 Ab (Figure 4D) and 211 Ab (Figure 5B,D).

The behavior of fluorescently labelled fibrils (with AF488 or/and AF647) was also reported in Cremades et al., 2012, where a thorough study of the effect of the fluorophores on the aggregation process and stability of the fibrils was performed.

Reviewer #3 (Remarks to the Author):

The authors have adequately addressed my comments.

We thank the reviewer for her/his positive comments on our work.

Reviewers' Comments:

Reviewer #2:

Remarks to the Author:

The authors have comprehensively addressed the comments raised, and included representative images of dyes, characterisation of cells, and more experimental details.

I have no further comments.

Reviewer #2 (Remarks to the Author)

The authors have comprehensively addressed the comments raised, and included representative images of dyes, characterisation of cells, and more experimental details.

I have no further comments.

We thank the reviewer for her/his positive comments on our work.